# Simulations suggest a constrictive force is required for Gram-negative bacterial cell division

Lam T. Nguyen[1], Catherine M. Oikonomou [1], H. Jane Ding[1], Mohammed Kaplan[1], Qing Yao[1], Yi-Wei Chang [1,3], Morgan Beeby[1,4] & Grant J. Jensen [1,2]

To divide, Gram-negative bacterial cells must remodel cell wall at the division site. It remains debated, however, whether this cell wall remodeling alone can drive membrane constriction, or if a constrictive force from the tubulin homolog FtsZ is required. Previously, we constructed software (REMODELER 1) to simulate cell wall remodeling during growth. Here, we expanded this software to explore cell wall division (REMODELER 2). We found that simply organizing cell wall synthesis complexes at the midcell is not sufficient to cause invagination, even with the implementation of a make-before-break mechanism, in which new hoops of cell wall are made inside the existing hoops before bonds are cleaved. Division can occur, however, when a constrictive force brings the midcell into a compressed state before new hoops of relaxed cell wall are incorporated between existing hoops. Adding a make-before-break mechanism drives division with a smaller constrictive force sufficient to bring the midcell into a relaxed, but not necessarily compressed, state.

[1] Division of Biology and Biological Engineering, California Institute of Technology, 1200 E. California Boulevard, Pasadena, CA 91125, USA. [2] Howard Hughes Medical Institute, 1200 E. California Boulevard, Pasadena, CA 91125, USA. [3] Present address: Department of Biochemistry and Biophysics, Perelman School of Medicine, University of Pennsylvania, 422 Curie Boulevard, Philadelphia, PA 19104, USA. [4] Present address: Department of Life Sciences, Imperial College London, South Kensington Campus, London SW7 2AZ, UK. Correspondence and requests for materials should be addressed to G.J.J. (email: jensen@caltech.edu)

Bacterial cells are protected from turgor pressure by a peptidoglycan (PG) cell wall that is composed of long glycan strands crosslinked by short peptides[1]. This relatively rigid sacculus allows cells to adopt specialized shapes, such as the rod shape of many Gram-negative bacteria. In order for the cell to change size or shape during growth and division, the pressurized sacculus must be carefully remodeled. This is accomplished by a set of cell wall remodeling enzymes including transglycosylases, transpeptidases, and endopeptidases. Experimental insights into the exact molecular mechanisms of these remodeling enzymes and how their functions are coordinated remain limited. Previously, we gained insight into these questions by building simulation software, REMODELER 1, to study cell wall synthesis during cell elongation[2]. In this software, a cylindrical cell wall is coarse-grained as chains of tetrasaccharide beads running circumferentially around the cylinder and connected by peptide crosslinks. The functions of transglycosylases, transpeptidases, and endopeptidases are explicitly modeled as beads. Using this software, we found that in order to maintain the integrity and rod shape of the cell, these remodeling enzymes have to coordinate with one another locally in synthetic complexes, but that no long-range coordination of the independent complexes is required. We also found that these complexes must contain a lytic transglycosylase to remove long, uncrosslinked glycan tails to clear the path for enzyme movement[2]. (Such an enzyme was independently identified experimentally[3].) How cells set their diameter over thousands of generations remains unclear, however. A recent study reported that the activities of the Rod system composed of RodA, class B penicillin-binding proteins (PBPs), and MreBCD reduce the diameter, while those of class A PBPs have the opposite effect[4]. It will be interesting to learn the molecular mechanisms of these two diameter-changing strategies and whether they alone set the cell diameter.

During cell elongation, the diameter of a rod-shaped cell is conserved. In contrast, during division, the diameter of the cell wall at the division site must become smaller and smaller. How the cell overcomes turgor pressure to remodel its cell wall to a smaller diameter remains unclear[5]. It is unlikely to be due to a fundamentally different mode of synthesis, since (a) partially overlapping and homologous sets of enzymes mediate remodeling in cell growth and division[6]; (b) these PG synthesis enzymes were shown to move around the cell's circumference during both elongation[7,8] and division[9,10]; and (c) in purified sacculi, glycan strands exhibit similar circumferential orientation throughout the length of the cell[11,12].

The protein FtsZ, a tubulin homolog found in nearly all bacteria and many archaea, forms filaments at the midcell during cell division[13–16]. It has been proposed that these filaments exert a constrictive force on the membrane and serve as a scaffold for the cell wall synthesis machinery[17]. Based on cryo-electron microscopy images of dividing cells, it has been proposed that GTP-hydrolyzing FtsZ filaments can generate a constrictive force either by switching conformation from straight to curved[14] or by overlapping to form a closed ring, which then tightens to constrict the membrane[15]. Alternatively, a recent study posited that FtsZ simply serves as a scaffold and that the constrictive force on the membrane is provided by the inward growing cell wall[18]. This model was suggested by the observation that the rate of inward cell wall growth is limited by the rate of cell wall synthesis, but not by the GTP hydrolysis rate of FtsZ.

In order to explore these different conceptual models, we modified our simulation software for the Gram-negative bacterial cell wall, REMODELER 1, to create REMODELER 2, which allowed us to test different mechanistic hypotheses of how inward cell wall growth might occur during division. We found that simply restricting the enzyme complexes to the midcell results in

elongation without constriction, even with a make-before-break mechanism of PG remodeling, suggesting that cell wall growth alone is not sufficient to drive Gram-negative bacterial cell division. We found that a constrictive force at the midcell results in cell wall division when the force is sufficiently large to initially constrict the midcell past the diameter of the unpressurized sacculus. If the constrictive force is slightly less, sufficient to constrict the midcell sacculus into a relaxed state, the addition of a make-before-break mechanism is now effective in facilitating division. These results are summarized in Supplementary Movie 1. Due to the difficulty of describing dynamic three-dimensional (3D) processes in words and static images, we recommend readers watch the movie in full before proceeding.

## Results

**Cell wall synthesis at the midcell.** To adapt REMODELER 1[2] into REMODELER 2 to study cell division, we added several features: (1) PG synthesis complexes were organized at the midcell, (2) a constrictive force could be implemented, and (3) the enzymes could build a multi-layered cell wall.

To simulate PG insertion during cell division, we built a starting PG sacculus and initiated four PG synthesis enzyme complexes randomly around the circumference within 10 nm of the midcell. To reduce the computational cost, we simulated a short cylindrical section (a midcell) of a miniaturized cell wall. Specifically, the starting PG cylinder was composed of 40 glycan hoops with each hoop consisting of 400 beads, representing 400 tetrasaccharides (Fig. 1a). The average strand length was set to be 14 tetrasaccharides, which is within the range of 11–16 tetrasaccharides reported experimentally[19,20]. As the distance between adjacent tetrasaccharides was $L_g = 2$ nm[2], the unpressurized PG cylinder had a radius of 127.5 nm (Fig. 1a). Under a turgor pressure $P_{tg} = 3$ atm, the cylinder expanded to a radius of 137.5 nm (Fig. 1b). We noticed that extension of the glycan strand was negligible due to its large linear rigidity (spring constant $k_g = 5.57$ nN/nm) and radius expansion was therefore mostly due to extension of peptide crosslinks that connected strand termini. Increasing the glycan strand length reduced the number of terminal crosslinks per hoop reducing the expansion of the cylinder (Supplementary Figure 1). To minimize any potential effects of changing the glycan strand length on the sacculus radius during the simulated cell wall remodeling, new glycan strands were also constrained to 14 tetrasaccharides on average when they were incorporated into the existing PG network. As in our previous simulations of PG remodeling[2], here we also assumed that the PG synthesis enzyme complexes insert new glycan strands in pairs (Supplementary Figure 2) and that the enzyme complexes can only act on PG substrates in their close vicinities, and cannot stretch the new pair of strands or pull them forward to crosslink them to distal peptides. At this stage of our model, we tested the hypothesis that simply organizing the PG synthesis complexes at the midcell can cause constriction[21,22]. In our simulations, however, insertion of new PG only elongated the cylinder without changing its radius (Fig. 1c, d), suggesting that additional factors are needed to induce division.

**PG remodeling under a constrictive force.** Since FtsZ filaments have been proposed to exert a constrictive force on the membrane, we implemented a constrictive force at the midcell to see if this allowed new PG to be incorporated in smaller hoops at the constriction site (see section "Methods/Constriction force"). Initially, the constrictive force made the midcell smaller before new PG was inserted (Fig. 2a). As new PG was inserted, further reduction of the midcell radius did not occur if $F_c$, the constrictive force divided by the sacculus circumference, was smaller than

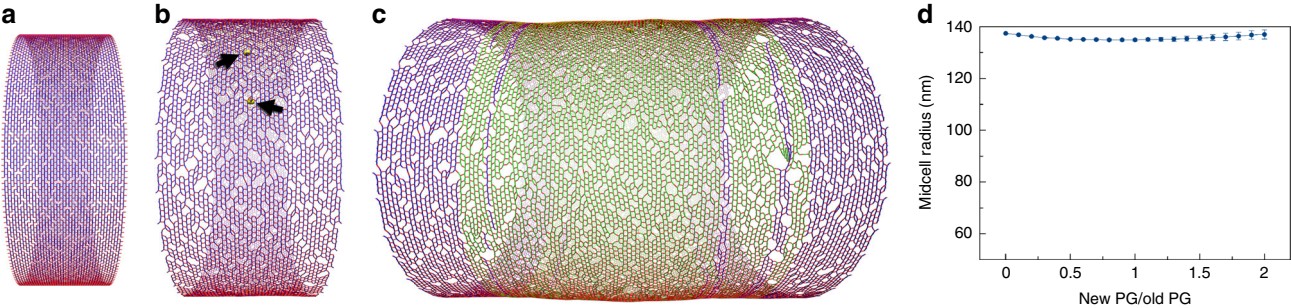

**Fig. 1** Cell wall synthesis at the midcell. Existing glycan strands are visualized in blue, new strands in green, and peptide crosslinks in red. The same color scheme is used for all other figures containing simulation snapshots. **a** A snapshot of the starting peptidoglycan (PG) cylinder in a relaxed state. The cylinder was composed of 40 hoops of glycan strands with 400 tetrasaccharides per hoop and had a relaxed radius of 127.5 nm. **b** A snapshot of the pressurized PG cylinder that was expanded to a radius of 137.5 nm under a turgor pressure of 3 atm. The arrows indicate PG synthesis enzyme complexes placed randomly at the midcell. **c** A snapshot of the PG cylinder after elongation to three times its original length by insertion of new PG. **d** The profile of the midcell radius with respect to the amount of new PG inserted, showing that PG synthesis at the midcell alone only elongated the cylinder without constricting it. The average of four simulations is shown. Error bars indicate standard deviation

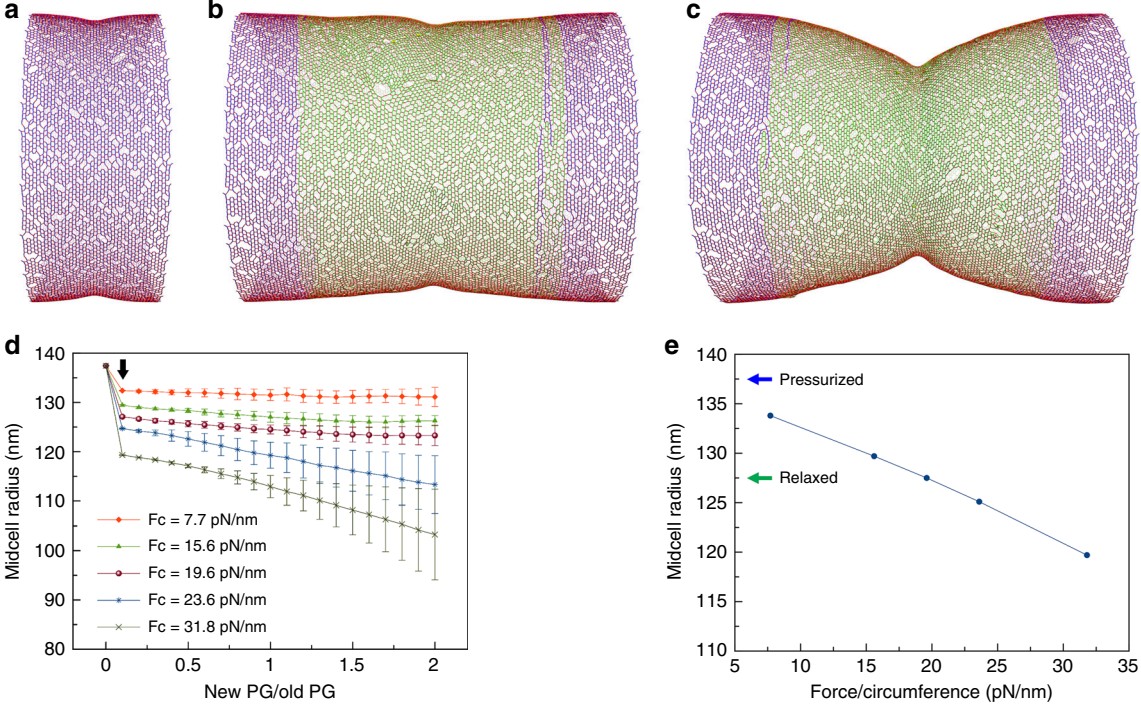

**Fig. 2** Cell wall synthesis under constrictive force. **a** A snapshot of the peptidoglycan (PG) cylinder showing that the force caused an initial constriction at the midcell before insertion of new PG. **b** A snapshot of the sacculus after new PG was inserted showing that when the force per circumference was $F_c = 15.6$ pN/nm, further constriction did not occur as new PG was inserted. **c** When $F_c = 31.8$ pN/nm, constriction occurred as new PG was inserted. **d** Profiles of the midcell radius with respect to the amount of new PG inserted show that PG insertion-induced constriction occurred only if $F_c$ was >19.6 pN/nm. The arrow indicates the initial constriction by the force before new PG insertion started. Each trace presents the average of four simulations. Error bars indicate standard deviation. **e** The dependence of the midcell radius on the force before new PG was inserted shows that the midcell became relaxed at $F_c = 19.6$ pN/nm. The blue arrow indicates the radius before the force was applied. The green arrow indicates the radius of a relaxed cell wall

20 pN/nm (Fig. 2b, d). The midcell did continue to reduce in size if $F_c$ was larger than 20 pN/nm (Fig. 2c, d). We found that at the transition point $F_c \sim 20$ pN/nm, the force initially constricted the midcell into a relaxed state where its radius was equivalent to that of an unpressurized cell, 127.5 nm (Fig. 2e). Therefore, a constrictive force alone can drive division if it is sufficiently large to initially bring the midcell into a compressed state, that is, reduce the midcell radius to less than that of a relaxed sacculus. Note also that our findings were limited to $F_c$ less than ~32 pN/nm since a larger force buckled the cell wall, making the simulation unstable. We speculate that such a buckling force scales with turgor pressure and cell radius.

We next analyzed in detail how a constrictive force might drive cell wall division. Without the constrictive force, the cell wall

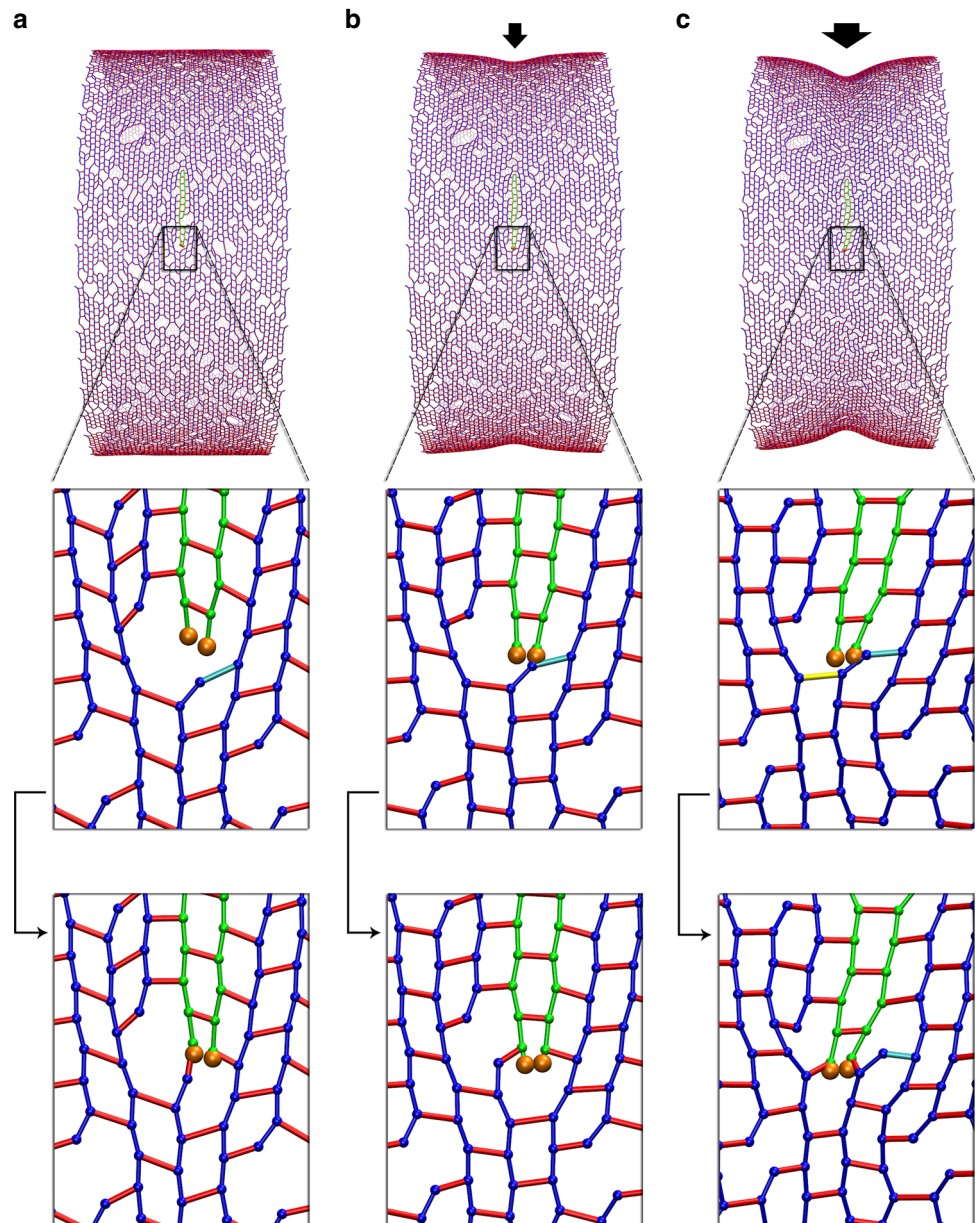

**Fig. 3** Detailed effect of the constrictive force. The arrows between the zoomed-in views in the bottom two rows indicate the time sequence of events. **a** Without the force, the new strand tips were at some distance from the existing crosslink that was targeted for cleavage (highlighted in cyan). After the target crosslink was cleaved, new peptidoglycan (PG) was matched one-to-one with the existing template. **b** In the presence of a small force, the midcell was squeezed, pulling the new strand tips closer to the target crosslink (highlighted in cyan). After the target crosslink was cleaved, the new PG was still matched one-to-one with the default template. **c** In the presence of a large force, the new strand tips were pulled past the default target crosslink (highlighted in cyan), therefore skipping it, and an upstream crosslink (highlighted in yellow) became the new target. After the new target crosslink was cleaved, the new PG was no longer matched one-to-one with the template. Note that several such template-skipping events occurred along each complete hoop of new PG

radius was maintained as new glycan beads were perfectly matched one-to-one with the existing template (Fig. 3a). Applying a small constrictive force ($F_c < 20$ pN/nm) squeezed the midcell and pulled the enzymes and the two new strand tips closer to the default target crosslink, but this did not interrupt the one-to-one template matching between the new beads and the existing beads and therefore did not reduce the midcell radius (Fig. 3b). On the other hand, in the presence of a large force ($F_c > 20$ pN/nm), the enzymes were pulled past the default target crosslink, skipping it and crosslinking the two new beads to a new

template that was upstream of the skipped template (Fig. 3c). Due to these skipping events, the two new PG hoops had fewer PG beads than the existing hoops, making the midcell radius smaller.

**PG remodeling under a make-before-break mechanism.** Next, we explored if and how cell wall growth alone could be sufficient to drive cell division without the presence of a constriction force. Conceptually, this can occur with a make-before-break mechanism, in which the cell wall synthesis machinery adds one or

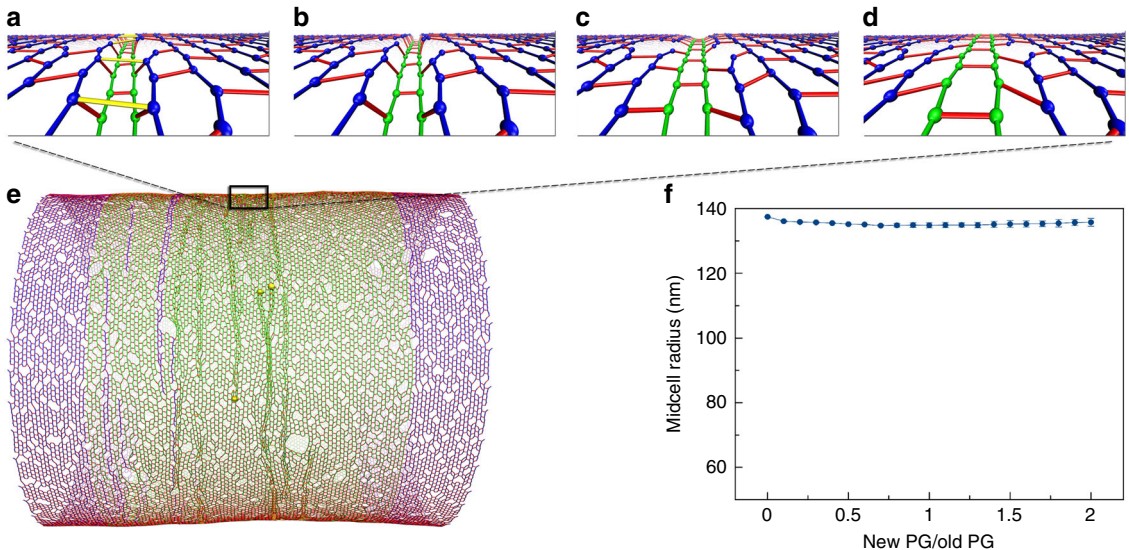

**Fig. 4** Cell wall synthesis with a make-before-break mechanism. **a–d** show the make-before-break mechanism occurring in sequence: **a** new cell wall was made underneath the existing network. The peptide crosslinks above the new peptidoglycan (PG) are highlighted in yellow. **b** The highlighted crosslinks were cleaved after complete hoops of new PG were made. **c**, **d** show the relaxation of the network after the cleavage event. **e** A snapshot of the PG cylinder shows that constriction did not occur with insertion of new PG. **f** The profile of the midcell radius with respect to the amount of new PG inserted. The averages of four simulations are shown. Error bars indicate standard deviation

several new PG layers that form a temporary septum underneath the existing PG layer (make) before hydrolases cleave the constraining peptide crosslinks above these new PG layers (break). If many layers are built in before any bond on the surface is hydrolyzed, the hoops of PG in the inner layer can potentially be made from fewer PG beads. While it is clear that Gram-positive bacteria divide by making a thick septum across the width of the cell, it has only recently been speculated that Gram-negative bacteria might also adopt this septation scheme, but with a thinner septum[23]. (For clarity, we use septum here to refer to the new PG layers beneath, but not including, the existing layer.) Upon examining our electron cryotomograms of dividing cells (see section "Can Gram-negative bacteria divide by a septation mechanism" for details), however, we concluded that if a thin septum exists at the dividing midcell, it must be thinner than ~4 nm, the resolution of the electron cryotomograms[24]. Accordingly, in our simulations, we limited septum thickness to one layer of PG and our findings are therefore limited to this assumption.

We simulated a make-before-break mechanism by decoupling PG synthesis (transglycosylation and transpeptidation) from PG hydrolysis (endopeptidation). Specifically, cleavage of existing peptide crosslinks was blocked until complete hoops of new glycan strands were crosslinked into the PG network underneath these crosslinks (Fig. 4a–d). Note that how this might occur at a molecular level remains unclear. The rate of endopeptidases was controlled so that only about one layer of new PG was present underneath the existing layer (see section "Methods/Make-before-break mechanism"). To mimic the volume exclusion effect between the outer and inner layers, before existing peptide crosslinks were cleaved, a repulsive force between these crosslinks and the new glycan strands was applied to separate them to a distance of $d_s = 2$ nm, the estimated thickness of one PG layer. To study the effect of the make-before-break mechanism alone, we did not apply a constrictive force.

Conceptually, if turgor pressure is not present, all the PG beads are evenly spaced at $L_g = 2$ nm (the length of one tetrasaccharide) on the same hoop. Matching the beads on the smaller hoop to

those on the larger hoop would then create mismatches (Supplementary Figure 3A). With a difference in radius $d_s = 2$ nm (the distance that separates the two hoops), the new hoop's circumference would be $2\pi d_s = 12.56$ nm shorter than that of the existing hoop. Since there are $N_b = 400$ beads on the existing hoop, the average mismatch per bead would be $\Delta s = 2\pi d_s/N_b$ ~0.0314 nm, which is small compared to the distance between the adjacent beads. However, if the first pair of beads on the two new strands (as discussed above, we assumed new strands are synthesized in pairs) are in register with their templates on the existing hoops (Supplementary Figure 3B), the second pair of new beads would be positioned ahead of their template by a distance $\Delta s = 0.0314$ nm. After ~30 pairs of beads are added to the new strands, the accumulated shifting of the strand tips would become $30\Delta s$ ~1 nm, about half the distance between adjacent beads (Supplementary Figure 3C). At this point reaching backward for the default crosslink template would become unfavorable, so instead the enzymes might skip the default template and reach forward for an upstream crosslink on the adjacent track (Supplementary Figure 3D). If such a skip occurred, the new strand tips would now trail their template by a distance of ~1 nm. It would then take another 30 beads for the new tips to catch up and once again be in register with their template (Supplementary Figure 3E). The cycle would then continue, leading to template skipping every 60 beads and a complete new hoop of only 394 beads, six beads less than the existing hoops. This would decrease the diameter of the midcell.

In the presence of turgor pressure, however, simulations of this make-before-break mechanism did not reduce the midcell radius (Fig. 4e, f). We found that once the existing crosslinks above the new PG hoops were cut, the inner hoops expanded to the size of the existing hoops (Fig. 4c, d), indicating that new hoops were made of a similar number of beads as existing hoops. Analyzing the problems, we found that turgor pressure expanded the cell wall making peptide crosslinks stretched and tilted away from the long axis of the cylinder (Fig. 5). As a result, beads were no longer evenly spaced on the same hoop. At breaks between glycan strands

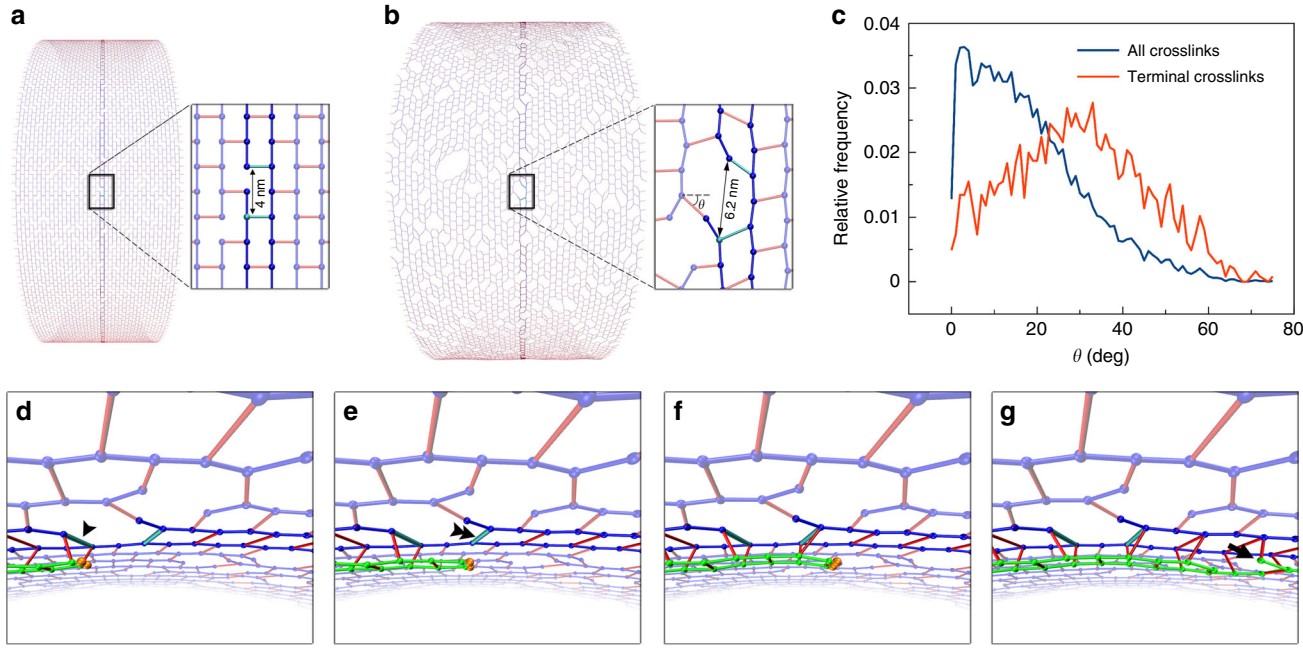

**Fig. 5** Effect of turgor pressure on the make-before-break mechanism. Two adjacent peptide crosslinks at a glycan break in a hoop are highlighted in cyan for visualization. **a** Without turgor pressure, the peptidoglycan (PG) was well ordered and the distance between the highlighted crosslinks was 4 nm. **b** Under turgor pressure, the cylinder expanded, peptides tilted, and the distance between the highlighted crosslinks increased to ~6.2 nm. $\theta$ depicts the tilting angle of a peptide crosslink. **c** Histogram of the tilting angles of all the peptide crosslinks (blue) and those connecting glycan termini (red). **d** An oblique view showing insertion of two new glycan strands in a make-before-break mechanism. When the new strand tips reach the first cyan-highlighted crosslink (indicated by the arrow head), they get ~0.4 nm ahead of their templates. **e** At the second cyan-highlighted crosslink (indicated by the double arrowhead), this small gain is offset by the additional 2.2 nm enlargement of the distance between the two crosslinks. **f** After the second highlighted crosslink, the strand tips fall behind their templates. **g** A break in the new glycan strands (indicated by the arrow) pulls the new strands forward, preventing them from falling behind their templates

in the hoop, the gap between the adjacent peptide crosslinks expanded from $2L_g = 4$ nm (Fig. 5a) to ~6.2 nm (Fig. 5b) as terminal peptides tilted an average of 30° (SD = 15°) (Fig. 5c). We observed that right before encountering a glycan break on the existing strands, which occurred every ~14 beads, the new strand tips had gotten ahead of their templates by an accumulated distance $s_a = 14 \ \Delta s = 0.44$ nm (Fig. 5d). At the glycan break, though, this small progress was more than offset by the 2.2-nm turgor pressure-induced expansion of the gap (Fig. 5e). At this stage, the new strand tips even fell behind their templates (Fig. 5f). This lag did not accumulate, however, because new strands also terminated, at which point the next new strands were pulled forward (Fig. 5g). This meant that template beads were not skipped, the new hoops had the same number of beads as the existing hoops, and constriction did not occur.

Since cell expansion due to turgor pressure varies with cell size (Supplementary Figure 4A), the difference in radius of the cell wall $\Delta r$ between a relaxed cell and a pressurized cell can be negligible for cells of sufficiently small sizes. For example, for cells with circumference of fewer than 150 tetrasaccharides (corresponding to a diameter of ~100 nm), $\Delta r$ would be smaller than $d_s = 2$ nm, which is the assumed thickness of one PG layer (Supplementary Figure 4B). In this case, a make-before-break mechanism could plausibly drive division in the absence of a constrictive force. In addition to cell width, turgor pressure may vary between cells. While most studies report turgor pressure in the range of 2–4 atm in Gram-negative bacteria[25–27], a turgor pressure as low as 0.3 atm has been reported[28]. At this low turgor pressure, cells of circumference smaller than 300 tetrasaccharides would expand <2 nm in radius (Supplementary Figure 4B). Still, for a make-

before-break mechanism to be effective in a cell the size of an *Escherichia coli* (~1500 tetrasaccharides in circumference), the turgor pressure would need to be on the order of 0.03 atm to make the radius expansion negligible (Supplementary Figure 4C).

**Make-before-break in the presence of a constrictive force.** Our observations suggested that in order for the make-before-break PG remodeling mechanism to be effective in constriction, the midcell must be in a relaxed state. We reasoned that a constrictive force squeezing the midcell could create this condition by restoring the gaps between adjacent peptide crosslinks at glycan breaks from ~6.2 nm to their relaxed size of ~4 nm (Fig. 6a). Implementing a constrictive force divided by circumference $F_c$ smaller than ~20pN/nm resulted in an initial constriction that did not completely relax the midcell. Predictably, in this scenario, insertion of new PG with a make-before-break mechanism did not cause further constriction (Fig. 6c). When $F_c$ was ≳20pN/nm, the midcell was completely relaxed or even squeezed to a smaller radius than the unpressurized cell (Fig. 2e). As we expected, in this condition make-before-break PG remodeling could now reduce the midcell radius (Fig. 6b, c). Note that while the constriction force alone could drive division if the magnitude of $F_c$ was larger than 20pN/nm, reducing the midcell to a compressed state (Fig. 2e), in the presence of the make-before-break mechanism, division started to occur at $F_c = 20$pN/nm since reducing the midcell to a relaxed state was sufficient.

**Can Gram-negative bacteria divide by a septation mechanism.** While Gram-positive bacteria divide by building a septum across

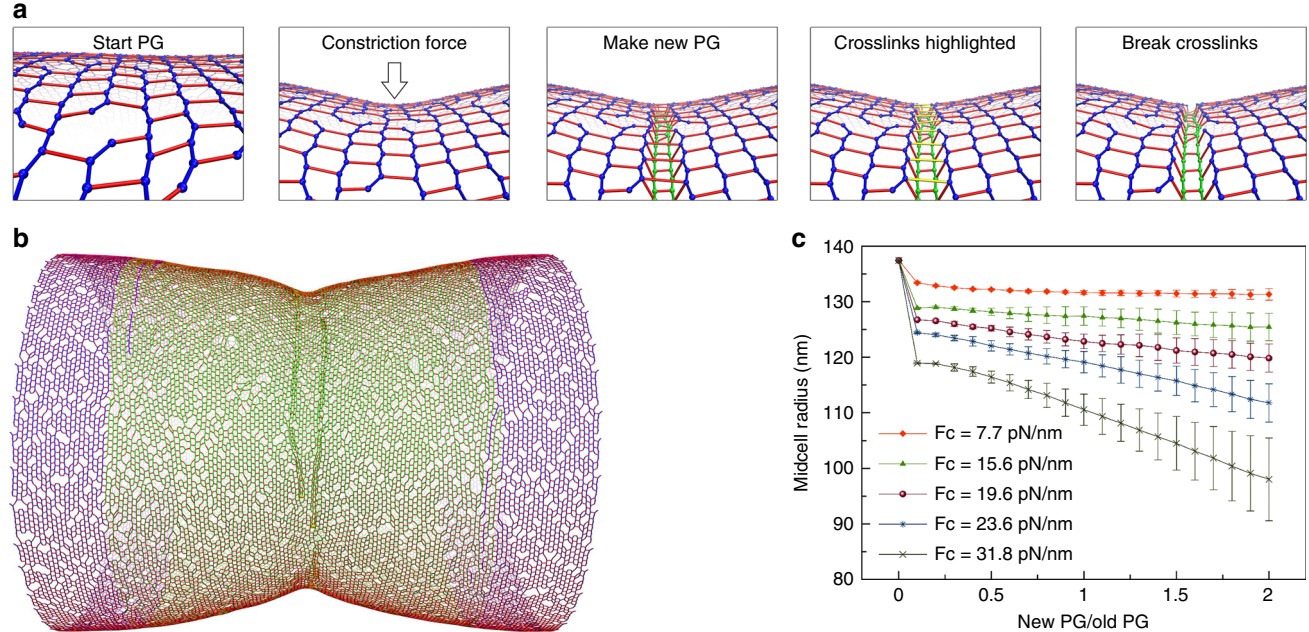

**Fig. 6** Cell wall synthesis in the presence of both a constrictive force and the make-before-break mechanism. **a** Representative snapshots of a simulation sequence: the constriction force caused an initial constriction, and new peptidoglycan (PG) was made underneath the existing network in complete hoops before crosslinks (highlighted in yellow) above the new hoops were cleaved. **b** A snapshot of the sacculus showing that constriction occurred upon insertion of new PG when the force per circumference $F_c = 23.6$ pN/nm. **c** Profiles of the midcell radius with respect to the inserted amount of new PG. Each trace shows the average of four simulations. Error bars indicate standard deviation

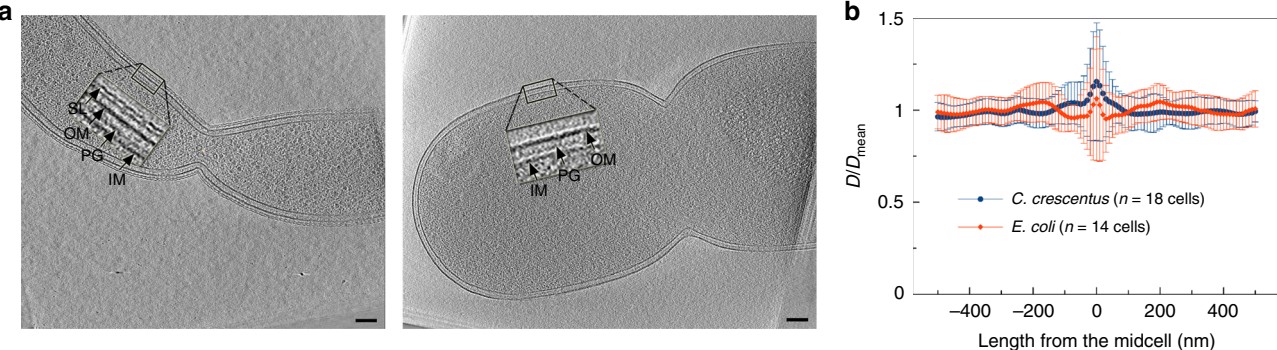

**Fig. 7** Electron cryotomography of dividing Gram-negative bacterial cells. **a** Representative tomographic slices through dividing *Caulobacter crescentus* (left) and *Escherichia coli* cells (right). Arrows point to the S-layer (SL), outer membrane (OM), peptidoglycan (PG), and inner membrane (IM). Scale bars indicate 100 nm. **b** Calculation of the distance $D$ between the two membranes shows that it is maintained throughout the invagination site at the midcell. Error bars indicate standard deviation

the cell, septation was observed in Gram-negative bacteria only when the cells were in abnormal conditions[29,30]. Erickson[23] speculated, however, that it is possible for Gram-negative bacteria to cheat turgor pressure and build a temporary septum if the periplasm and cytoplasm are iso-osmolar[23]. Based on published electron tomograms of dividing Gram-negative bacteria, the author suggested such a septum might be built while hydrolases actively degrade outer layers of the advancing septum, keeping it thin. To determine if such a septum exists in dividing Gram-negative bacterial cells, we carefully examined 3D electron cryotomograms collected in our lab of intact frozen-hydrated cells of six species: *Caulobacter crescentus*, *Escherichia coli*, *Proteus mirabilis*, *Myxococcus xanthus*, *Cupriavidus necator*, and *Shewanella*

*oneidensis*. In all cases, we could not discern any thickening of the wall at the dividing midcell that might indicate the existence of a thin septum (Fig. 7a; Supplementary Figure 5). We also observed that the distance between the inner and outer membranes remained constant throughout the midcell (Fig. 7b). We therefore concluded that if a thin septum exists at the dividing midcell, it must be thinner than ~4 nm, the resolution of the electron cryotomograms[24].

## Discussion
Together, our results demonstrate how 3D modeling of molecular details can provide insights into complex symmetry-breaking processes such as cell division. By simulating how rod-shaped

Gram-negative bacteria could divide their cell walls, and within the assumptions of our model, we found that a constrictive force is the key factor driving constriction, while cell wall remodeling by a make-before-break mechanism can only facilitate the process. Our work advances understanding of how the molecular consequences of the make-before-break idea play out on a large scale, beyond individual glycan strands as originally described[31,32].

Many factors involved in cell division remain unknown, however, and our model is a simple coarse-grained model that does not represent any particular bacterial species. It is a theoretical, simplified construct meant to explore the fundamental mechanistics ideas in the field. Future experiments will surely further clarify the situation. For example, the PG synthesis mode during cell division and that during cell elongation might be completely different, resulting in different architectures of cell wall, for example, different glycan strand length distributions. While the two major bifunctional transglycosylase/transpeptidase enzymes PBP1A and PBP1B of *E. coli* showed some overlapping roles[33], the two monofunctional transpeptidases PBP2 and PBP3 (also known as FtsI), which are essential for cell elongation and cell division, respectively, are likely part of two different complexes, namely, an MreB-associated PG synthesis complex[34] and an FtsZ-associated PG synthesis complex[10]. While during cell elongation, the circumferential movement of MreB was shown to be driven by PG synthesis and dependent on the activities of PBP2[35], during cell division, the speed of PBP3 was shown to depend on the treadmilling rate of FtsZ[10].

To decrease the radius of the midcell, the cell needs to add smaller and smaller hoops of new PG. Conceptually, this can occur by at least three mechanisms. (1) Because existing PG hoops are stretched by turgor pressure, if by some unknown force new PG hoops are stretched even further at the time they are incorporated between existing hoops, the new hoops would have fewer PG beads and therefore become smaller as the system relaxes. (2) If a mechanism exists to initially compress existing hoops at the midcell to become even smaller than relaxed hoops and new, relaxed PG hoops are incorporated between these existing hoops, the new hoops would have fewer beads and remain small once the compressing force terminated. (3) If a mechanism exists to initially relax existing hoops at the midcell and new, also relaxed, PG hoops are made inside (at smaller radius—i.e., make-before-break) existing PG hoops, the new hoops would again have fewer beads. Note that in all three models, new PG and old PG are brought to a similar stretching degree before the former is inserted into the wall. Here we showed that Model 2 is plausible with a constriction force alone and Model 3 is plausible with a combination of a constriction force and a make-before-break mechanism. We did not simulate Model 1 as we judge it unlikely to occur in real cells. Nevertheless, we cannot currently rule out the possibility that an unknown force pulls the enzymes forward, stretching the new glycan strands before incorporating them. Such a PG pre-stretching mechanism would enable cell division with a much smaller constrictive force[36] or in a combination with a make-before-break mechanism perhaps render a constrictive force unnecessary. Likewise, we cannot rule out the possibility that cells might divide by a completely different mechanism that has not yet been discussed in the literature. For example, while FtsZ treadmilling has been speculated to help distribute cell wall material evenly around the cell's circumference[9,10], such a directional movement of long filaments might cause constriction in an unknown mechanism.

What is the role of a constrictive force? FtsZ filaments have been shown to be able to constrict liposomes[37], but it is unclear if they exert a constrictive force on the membrane in vivo and if this force is required for cell division. Here we found in silico that to make new PG hoops smaller than existing hoops, the midcell

needs to be initially constricted to at least a relaxed state, with further constriction occurring only after new PG is inserted. In this model, the constriction rate is limited by the slower of either the force generator (presumably FtsZ) or the PG synthesis rate. Therefore, the finding by Coltharp et al.[18] that the inward growth rate of the cell wall is limited by the rate of PG synthesis, but did not change even when the GTP hydrolysis rate of FtsZ was reduced 90%, might simply reflect that the PG synthesis rate is much slower than the action of FtsZ. Indeed, it has been reported that an FtsZ mutant with a GTP hydrolysis rate 3% that of wild-type FtsZ resulted in very slow growth of colonies[38].

In our model, the total initial constriction force needed to be at least ~15 nN (corresponding to a force divided by circumference $F_c$ ~20pN/nm) to enable cell wall division. Assuming that the constrictive force is generated by FtsZ, each monomer of which has been estimated by molecular dynamics simulations to generate 30 pN[39], our estimated force is equivalent to the action of 500 FtsZ monomers, which could form a continuous filament ~2.2 μm long or 15 filaments of an average length of 150 nm. This is reasonable, considering an estimated ~5000–7000 FtsZ molecules per cell measured in *E. coli*[17] and the fact that our simulated sacculus is a third the size of an *E. coli* cell. Note that it has recently been speculated that excess membrane synthesis might also generate a constrictive force[5].

Can cell wall growth alone drive constriction? While cell wall growth has been speculated to partially or primarily drive constriction during cell division[18,21], our simulations showed that cell wall growth via a make-before-break mechanism failed to cause cell division in the absence of a constriction force. We found that for the make-before-break mechanism to have an effect, the new PG hoops must be made inside relaxed existing hoops and therefore a constriction force is needed to initially relax the midcell. In theory, the need for a constriction force could be bypassed if the enzymes could make a multi-layered septum. Once the septum thickness was equal to or larger than the difference in radius between the pressurized cell and the relaxed cell, the innermost layer of the septum would be in a relaxed state, allowing the next PG layer to be made of hoops containing fewer PG beads (Supplementary Figure 6). For this scenario to occur, in the case of our modeled cell wall, which had a radius of 127.5 nm when relaxed and 137.5 nm when pressurized, the septum would have to contain at least five PG layers (assuming each layer is 2 nm thick). By this logic, a cell the size of an *E. coli*, whose circumference is ~1500 tetrasaccharides, would need a septum ~65 nm thick for this mechanism to drive division (Supplementary Figure 4A–B). However, we saw no such septa in our electron cryotomograms of dividing cells. We therefore think it unlikely that this mechanism is the primary driver of cell division.

## Methods
**Simulation of cell wall synthesis**. Here we only briefly describe our simulation system. For a more detailed description, please see our previous papers[2,40]. The source code of our simulations is provided with the manuscript as Supplementary Software.

**Cell wall**. We chose force constants for our coarse-grained glycan strand and peptide crosslink model to recapitulate the results of atomistic simulations[2]. We coarse-grained the cell wall such that each glycan strand is represented as a chain of beads, each bead represents one tetrasaccharide, and the peptides attached to the beads alternate between the left and right sides. Note that previously, we started with models of cell elongation in which alternation of peptides on the two sides of the glycan strand was not implemented, allowing formation of successive crosslinks on the same side[2]. These models resulted in cell bulging since short segments of one strand could be crosslinked to long segments of another strand. Considering that adjacent disaccharides rotate roughly 90° along the strand[41–44], we then implemented alternation of successive peptides on the two sides of the strand and this helped maintain the cell's rod shape. Adjacent glycan beads are connected by springs of a relaxed length $l_g = 2$ nm and a spring constant $k_g = 5.57$ nN/nm. The bending stiffness of the strand is $k_b = 8.36 \times 10^{-20}$ J and the relaxed angle at the

beads is $\theta_0 = 3.14$ rad. We modeled peptide crosslinks as worm-like chains such that if the peptide end-to-end extension $x$ is larger than $x_0 = 1.0$ nm, the following force is applied:

$$F(x) = k_{WLC}\left[\frac{L_c - x_0}{4\{1 - (x - x_0)/(L_c - x_0)\}^2} - \frac{L_c - x_0}{4} + x - x_0\right], \quad (1)$$

where $L_c = 4.8$ nm is the contour length of the peptide crosslink and $k_{WLC} = 15.0$ pN/nm is the force constant.

To build the initial cell wall cylinder, glycan strands were placed along circumferential hoops of the same radius, then opposing peptides on adjacent hoops were connected to form crosslinks[40]. Previously, in order to reduce the computational cost, most of our simulations started with a small sacculus with a circumference composed of 100 tetrasaccharides[2]. In the current simulations, to allow the midcell radius to constrict over time, we used a starting sacculus with a circumference of 400 tetrasaccharides. To reduce the computational cost, since PG remodeling only occurs at the midcell during cell division, we removed the two caps of the starting sacculus and built a cylinder only 40 glycan hoops wide.

**PG remodeling enzymes**. To model enzyme activities, we assumed they function in complexes[2,40]. Four enzyme complexes were added at the midcell. In each complex, three types of PG remodeling enzymes are explicitly represented as beads and a house-keeping enzyme that cleaves the long tails of glycan strands is implicitly implemented. Specifically, there are two transglycosylases that each synthesizes a glycan strand (so two strands emerge from the complex) (Supplementary Figure 1). On average, each transglycosylase adds a tetrasaccharide bead every $10^3$ time steps. Transglycosylase then translocates to the strand tip to be ready to add another bead. Note that previously we hypothesized that transpeptidation facilitates translocation. Specifically, the probability of translocation is once every $2 \times 10^6$ time steps if the last-added bead is not crosslinked, but this probability becomes once every $3 \times 10^4$ time steps after the last-added bead is crosslinked (these numbers were arbitrarily chosen because we were not aware of experimentally reported enzyme rates)[2]. Considering that the modeled sacculi in our current simulations were four times larger than those in our previous simulations, to speed up the current simulations, we increased the probability of transpeptidation-facilitated translocation 10 times to become once every $3 \times 10^3$ time steps. To maintain an average glycan strand length of 14 tetrasaccharides, the termination probability of strand elongation is also increased two-fold to once every $2 \times 10^6$ time steps. Note also that in our previous simulations, interactions between transglycosylases and outer membrane lipoproteins LpoA and LpoB were implemented that prevented the transglycosylase–lipoprotein complex from crossing through glycan strands or peptide crosslinks. To enable the make-before-break mechanism, these transglycosylase–lipoprotein interactions were removed from the current model, allowing transglycosylases to freely move across strands and crosslinks.

One endopeptidase exists in each enzyme complex to cleave existing crosslinks. In our previous simulations, when an endopeptidase diffused across a crosslink, the enzyme cleaved the crosslink with a probability of 0.1. To speed up our current simulations, every 10 time steps, if the distance from the endopeptidase to a crosslink is within 3.0 nm, the enzyme captures then cleaves the crosslink. If there are multiple crosslinks within this reaction distance, the probability of crosslink $i$ being chosen is calculated as

$$P_i = \frac{1/d_i^2}{\sum^1 /d_i^2}, \quad (2)$$

where $d_i$ is the distance from the endopeptidase to crosslink $i$.

There are three transpeptidases in each complex, one crosslinking the two new strands to one another and the other two crosslinking the pair to the existing network (Supplementary Figure 1). Previously, the probability of a transpeptidase capturing a peptide of a PG bead at a distance $d$ was given as $P_{tp} = (1 - d/d_0)^2$, where $d_0 = 2.0$ nm was the reaction distance. To speed up the current simulations, $d_0$ is increased to 3.0 nm. Note that increasing the modeled rates of the enzymes did not change the principles driving cell wall remodeling in our simulations.

To constrain the enzymes into a complex, they were tethered such that a force, $F_{ez} = -k_{ez}(d_{ez} - D_0)$, was applied to draw two enzymes closer together if their distance $d_{ez}$ became larger than $D_0 = 1.0$ nm. The force constant was chosen to be $k_{ez} = 10$ pN/nm. Since in real cells, enzymes are in the periplasm, we introduced a force normal to the surface, $F_{surf} = -k_{surf}(d_s - d_{s0})$, on an enzyme if it moved a distance $d_s$ larger than $d_{s0} = 3.0$ nm away from the surface. The force constant $k_{surf}$ was chosen to be 500 pN/nm.

To model enzyme diffusion, a random force was applied to each enzyme every time step. Using the Box–Muller transformation[45], two random numbers from a Gaussian distribution were generated as:

$$r_1 = \cos(2\pi u_2)\sqrt{-2\ln(u_1)} \quad (3)$$

and

$$r_2 = \sin(2\pi u_2)\sqrt{-2\ln(u_1)}, \quad (4)$$

where $u_1$ and $u_2$ are two random numbers from a uniform 0–1 distribution. For $N$ enzymes, $3N$ Gaussian random numbers were generated and then scaled by a force constant of 500 pN to obtain $3N$ Cartesian components of the random forces.

**Turgor pressure**. To calculate the effect of turgor pressure, we first calculated the volume $V$ enclosed by the pressurized sacculus. To do this, the surface of the sacculus was divided into a series of polygons whose vertices were at positions of the PG beads. Each polygon was further divided into a set of triangles, and each triangle together with the center point of the sacculus formed a tetrahedron[40]. In the end, the sum of the tetrahedrons' total volume $V = \sum V_i$, where $V_i$ was the volume of tetrahedron $i$.

The force on the sacculus was then calculated as $F_{tg} = -\nabla E_{vol}$, where $E_{vol} = -P_{tg}V$ is the work done by turgor pressure to inflate the sacculus. Specifically, the force on bead $j$ was calculated as following:

$$F_j = P_{tg}\sum \nabla_j V_i. \quad (5)$$

As in our previous simulations, turgor pressure was chosen to be $P_{tg} = 3.0$ atm.

**Constriction force**. When a constriction force is applied at the midcell, it creates an inward pressure $P_c$ satisfying

$$F_c = \int P_c dx_m, \quad (6)$$

where $F_c$ is the constriction force divided by the circumference and $x_m$ is the distance from the midcell. To model the force, if the absolute value of $x_m$ is <50 nm, a constriction pressure $P_c(x_m) = P_0 e^{-x_m^2/\sigma^2}$ is applied, where $\sigma = 10$ nm and $P_0$ is calculated using the following equation:

$$P_0 = \frac{F_c}{\int e^{-x_m^2/\sigma^2} dx_m}. \quad (7)$$

**Make-before-break mechanism**. To implement the make-before-break mechanism, existing peptide crosslinks are marked as "constraining crosslinks" once new glycan strands are formed underneath the crosslinks. Cleavage of a constraining crosslink is delayed for $10^6$ time steps. After that, cleavage can occur by four scenarios: (a) an endopeptidase is within 2 nm of the constraining crosslink, (b) two PG layers exist beneath the crosslink, (c) a random cleavage with a probability of once every $10^7$ time steps, or (d) the constraining crosslink has existed for $10^9$ time steps. We found that these probabilities resulted in a septum ~1 PG layer thick.

In real cells, different PG layers would be separated due to the volume exclusion effect. To mimic this effect in our model, if a glycan strand is underneath a constraining crosslink and their separation $d$ is less than the thickness of one PG layer $d_s = 2.0$ nm, a repulsive force $F_r = k_r(d_s - d)$ is exerted on the crosslink and the two PG beads that are closest to the crosslink, where the force constant $k_r$ was arbitrarily chosen to be 200 pN/nm.

**Langevin dynamics**. The system was evolved using Langevin dynamics such that coordinate $X(t)$ of a bead of mass $m$ and interaction potential $U$ was calculated as

$$m\frac{d^2X}{dt^2} = -\nabla U(X) - \gamma\frac{dX}{dt} + R(t), \quad (8)$$

where $\gamma$ is damping constant and $R$ the random force on the bead. For simplicity, we assumed the inertia of the bead was negligible, making the displacement $dX$ a linear function of the total force $F_{tot}$ on the bead as

$$dX = \frac{1}{\gamma}[-\nabla U(X) + R]dt = \frac{1}{\gamma}F_{tot}dt. \quad (9)$$

To prevent system instability, the maximum displacement of the PG beads, which corresponds to the maximum force $F_{max}$, was set to be $d_{max} = 0.01$ nm. Displacement of bead $i$ was then calculated as

$$d_i = d_{max}\frac{\gamma_{max}F_{tot}^i}{\gamma_i F_{max}}. \quad (10)$$

**Electron cryotomography**. Bacterial strains were grown and imaged as described: *Caulobacter crescentus*[16]; *Escherichia coli*[46]; *Myxococcus xanthus*[47]; *Cupriavidus necator*[48]; *Shewanella oneidensis*[49].

**Measurement of the distance between the inner and outer membranes**. A tomographic slice 10 nm thick through a central plane along the long axis of the cell was captured using the IMOD software[50]. Each membrane (inner and outer) was manually traced and represented by a set of points evenly spaced along the line. This process was repeated for the membranes on the opposite side of the cell. The location of the midcell was determined by the shortest distance between the two traces of the inner membrane. The distance between the inner and outer membrane on each side was then calculated for points up to 500 nm from the midcell in both directions.

**Reporting summary**. Further information on experimental design is available in the Nature Research Reporting Summary linked to this article.

## Code availability
The source code of our simulations is provided as Supplementary Software.

## Data availability
The data supporting the findings of this study are available within the paper and its Supplementary Information files.

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

## Acknowledgements
We thank Martin Pilhofer for sharing the electron cryotomogram of an *E. coli* cell shown in Fig. 4, Debnath Ghosal for helpful discussions, and Andrew Jewett for assisting with membrane-tracing software. This work was supported by the National Institutes of Health (grant R35 GM122588 to G.J.J.).

## Author contributions
L.T.N. designed and ran simulations and wrote the manuscript. C.M.O. wrote the manuscript. H.J.D. ran simulations and did analysis of electron cryotomograms. M.K., Q.Y., Y.-W.C. and M.B. performed electron cryotomography. G.J.J. designed simulations and wrote the manuscript.

## Additional information

**Competing interests:** The authors declare no competing interests.

