## [Peer Review File · Nature Communications]

Reviewers' comments:

Reviewer #1 (Remarks to the Author):

Simulations suggest a constrictive force is required for Gram-negative bacterial division

In this Manuscript, Nguyen et al investigate the mechanisms allowing the constriction of Gram-negative bacterial during division.

To this end, the authors rely mostly on simulations of the cell wall mechanics, in combination with analysis of electron microscopy images.

They report that the "make before break" mechanism of wall generation does not suffice to create a division furrow, while this is permitted by a constriction force, that could be attributed to FtsZ contraction.

This work is interesting as the septum-less division in Gram-negative bacteria is an interesting feature, contrasting with the usual view from Gram-positive bacteria, yeast, and plants. The manuscript builds from the 2015 article introducing the simulation methods for wall growth, and extends it to study cell division. The exact mechanism of force generation is not within the scope of this manuscript.

The concerns raised by this work are mostly the extreme specificity of the simulation software, and the lack of more general physical considerations.

Specific comments :

1- Simulation

Simulating bacterial cell wall mechanics is no small feat. The exact assumptions of the model, and its implementation, would greatly benefit to be better detailed. Such a detailed description is also sadly incomplete in the 2015 PNAS article.

1-A : Physics

- Limitation of steady-state cell radius : except from the initial number of beads per hoop, what sets the cell radius ? If it is indeed only the initial number of beads per hoops, errors in insertion would make the cell diameter unstable : the larger the cell, the more the pressure force ($P R^2$). Over thousands of generations, there must exist a physical or biochemical basis maintaining cell diameter.

While the current study addresses a single generation, this is crucial to understand the physics underlying the simulations, and therefore its accuracy.

- Strands as lines vs ribbons : in the article, strands are lines of beads. When the authors mention a specific angle $\theta_0=3.14$ between segments, may we safely assume they mean a zero spontaneous curvature for the strand ? Note that for a line, spontaneous curvature is not well defined, it is so only for ribbons, that have an orientation.

In the model, the crosslinks alternate between each side of the strand. Once again, this amounts as treating the strand as a ribbon rather than a line. As far as I understood, this alternative pattern was set directly in the simulation (by preventing successive binding between two same strands), not derived from an underlying mechanisms. This is a very strong assumption. How much does this affect the stability of the cell radius ? Even a few errors would likely create the cell wall to be unstable at some timescale (see first point).

- Length of strands and speed of enzyme translocation is also given by simulation assumptions and does not come from physical or biochemical principles. This could be problematic if there were feedbacks between strand length and mechanics. Here this should not play too large of a role, but it would be useful to confirm.

- The authors assume that strand extension depends on crosslinking. If I understood correctly, extension is 3000x more likely after crosslinking. This is a crucial assumption both for the stability of cell radius and the make before break mechanism.

1-B : Simulations

- Compilation : the provided code could not be compiled out of the box as suggested in the readme. I had to fix an overlong line (132 characters), replace the instruction `-openmp` by `fopenmp` and add the instruction `-fno-check-range` (gfortran on Ubuntu 16LTS). It then compiled and run on 4 cores in parallel.

- Open source code : the authors provide the source code of `remodeler2` which is a great choice. The parameter file has a simple syntax and the parameters can be modified easily. Due to limited commenting, modularity (limited partly by the choice of Fortran90) and documentation, it should not be considered as a large-purpose platform that would be widely used and built upon by the community. Rather, it is a program fulfilling a specific simulation task in a fast and efficient way.

2- Constriction force

The constriction force per unit length (a tension, maybe this term could be used to avoid confusion) necessary is of the order of 20pN/nm. The buckling tension is of the order of 35pN/nm. It is important to highlight that these values are expected from scaling arguments (tension~P*R) and should not be overly model dependent. This scaling with size and pressure could be confirmed by simulations ; the numerical cost of simulations (not discussed in the article) might be a limiting factor for this systematic approach.

The tension of 20pN/nm, i.e. a global force of 15nN is the reason why other turgid walled cell need to form a septum. The absence of a such septum is the crucial point addressed by this article. The conclusion that make before break (MBB) cannot work because of the wall tension caused by pressure is very important.

However, the authors quickly dismiss the hypothesis that would allow MBB to work, namely the existence of a pre-stress. For instance, a pre-stress has been reported for spectrin in red blood cells (Turlier et al. Nat Phys 2016). Could FtsZ force cause a such pre-stress ? What value should we expect for a pre-stress, so that MBB works ? It is likely that this value is similar to the constriction tension.

Lastly, a tension of ~20pN/nm matches, or is beyond, the usual estimate of membrane-breaking tension. What prevents here the membrane from (a) peeling of the wall (b) rupturing ?

Reviewer #2 (Remarks to the Author):

This study builds on the authors' 2015 paper, which derived a computer model for PG assembly in Gram-negative bacteria. That model, based on a PG wall that is one layer thick, successfully predicted elongation of the PG wall, avoiding bulges and ruptures and maintaining a constant diameter over generations. Now the authors adapt their model to try to explain how PG remodeling can achieve constriction during cell division. This addresses a current controversy in the field of bacterial cell division: can PG remodeling provide a constriction force by pushing the membrane from the outside, or is the constriction force provided primarily by FtsZ, with PG remodeling just accommodating constriction? The authors try several schemes for insertion of new strands, and conclude that two mechanisms can accommodate constriction of the PG wall. A major conclusion is that constriction against turgor only occurs when there is an external force that can constrict the sacculus to a radius less than that of a relaxed (no turgor) state. Therefore PG remodeling is not driving constriction, but simply accommodates constriction driven by the

external force, likely FtsZ. This will not be the last word on the controversy – its conclusions are limited to the assumptions of the model. But advocates of PG pushing will be challenged to come up with a molecular mechanism, perhaps with modified rules for the network of glycan strands and peptide crosslinks and the insertion of new ones. I recommend publication after attention to some questions and concerns.

1. The model is based on a PG wall limited to one layer. Is there evidence that the PG is really only one layer, as opposed to two or three layers?
2. The cryotomograms are introduced in the middle of the section “make-before-break mechanism, interrupting the flow of that section.” I would suggest moving the cryoEM to a separate section at the end “Can Gram-negative bacteria divide by a septation mechanism.” Here you might expand a bit and explain that you are addressing Erickson’s (2017 Bioessays) suggestion, that a septation mechanism could allow cell division to cheat turgor pressure by building a new septum inside a periplasm that was isoosmolar with the cytoplasm. However, this mechanism requires at least a vestigial septum protruding from the cell wall into the periplasm. Erickson suggested that this might exist based on published cryotomograms. In contrast, you are now presenting an expanded gallery of new cryotomograms that show no hint of a septum at your 4 nm resolution. You might clarify that this failure to see a septum is presented as a contradiction of his hypothesis, as applied to Gram-negative bacteria.
3. Keep in mind, however, that Gram-negative bacteria can grow PG septa under some conditions (Erickson, 2017; Uehara et al, J. Bacteriol. 191:5094). This suggests that the purely constrictive division seen in these cryotomograms may share mechanisms with septal division. This would require a more complex system than the present model.
4. Fig. 3 is difficult to interpret. The text speaks of beads, which are not on the Fig. From the previous paper I believe beads should be at the intersection of each peptide with its glycan. It seems that the cyan target peptide is changed from B to C: the one in C is now on the left and is a bit further down. It may help to call attention to this detail. Also in Fig. 6 I could not find the highlighted strands. If they are orange I can’t distinguish them from red.
5. In Discussion the authors write “(2) If a mechanism exists to initially compress existing hoops ..(3) If a mechanism exists to initially relax existing hoops.” I think compress and relax may refer to the same thing: constricting to the same state as a relaxed sacculus? Do clarify.
6. I think the “make before break” mechanism is described in a confusing order (apart from the interruption by the cryotomograms). I suggest first outlining how it might work, by the forward skipping mechanism in Fig. S3. Then explain that this would actually work without turgor. Then explain that turgor increases the gaps between glycans, and this defeats the skipping mechanism. Am I correct that the defeat is primarily due to the average length of the glycan strand being about the same as the distance needed for a skip.
7. A recent paper from Rojas..Huang et al (Nature 2018) showed that the outer membrane contributed substantially to the mechanical rigidity of the cell wall. Does this affect your model?
8. The forces you find necessary are very large, much larger than those previously suggested by Lan..Sun (2007). Can you discuss how your models differ from Sun’s? Also, the forces given here are for the 137 nm radius. If you move to the 500 nm radius of E. coli, do the forces get larger or smaller?

Harold Erickson

Reviewer #3 (Remarks to the Author):

This paper addresses one of the most important questions in bacterial cell biology: how do cells generate the forces required to divide? Currently, there are two schools of thought to explain the mechanism of bacterial cell division: first, that the bacterial tubulin FtsZ is the main force generator providing the an inward directed force required for the cell to constrict and split into two and second, that the force originates from the other side of the membrane, namely by the cell wall that is growing inward. According to the later model, the cytoskeletal ring provided by FtsZ would

only act as scaffold guiding the inward growth of the cell septum. Using a very elegant a powerful previously established computer simulation approach, the authors of this paper were able to test both models "in silico". Their study nicely and convincingly shows that for cell constriction a combination of both is most probably applied: inward growth by a make-before-break-mechanism supported by an inward pulling force compensating for the turgor pressure.

The claims of the paper are of interest for many other researcher in the community, even though the hypotheses made have existed for a long time and the findings are probably not too surprising. Nevertheless it will help to motivate and design future experimental work.

Below are some concerns I have with the paper, I hope they are helpful to either clarify or improve the manuscript.

The main assumption of this study and its model is that cell wall synthesis does not work fundamentally different during elongation and division with the only difference being an inward directed force during cell division presumably provided by the Z-ring. The author's provide three reasons to justify this assumption, however, none of them is actually very convincing and would need a more detailed explanation .

1. Enzymes involved in elongation and division have overlapping roles: While this is true for PBP1B and PBP1A, other proteins are essential and only perform one task, such as PBP2 (elongation) and PBP3 (cell division). It is true that they are structurally and functionally related to other PG synthases, however, they are part of very different complexes and subject to different regulation. Therefore cell division complexes might function very differently than elongation complexes and could give rise to a different connectivity of the PG network. This was also pointed out by Turner et al 2018 cited in this paper: "the poles and the cylindrical part of the cell are biosynthesised by different sets of nano-machinery"

2. PG synthesis proteins were shown to move around the cell's circumference during both elongation and division: This over-simplifies what we currently know about the intracellular dynamics of PG synthases and their activities. First, in the case of E. coli elongation, directional movement was only found for filaments of MreB, which was dependent on the elongation specific TPase PBP2, not for the division specific TPase PBP3 or GT-TPases PBP1A and PBP1B. In contrast to FtsZ, which itself treadmills to drive the motion of PBP3. Accordingly, the PG synthase complex are either the driver (elongation) or the passenger (division) of the observed motion, which suggests that cell wall synthesis for both processes could be very different. In fact, it is currently not known if during septal PG synthesis the corresponding enzymes are moving (as PBP3), static (in complex with a cytoskeletal element) or diffusing (freely diffusing, see Cho et al 2017). Therefore, the situation is much more complex and should be appropriately discussed.

3. The orientation of glycan strands is similar at the poles and cylindrical parts of E. coli: this was nicely shown experimentally by Turner et al 2018, but even if the orientation of strands is similar, the length distribution might be very different.

Next, the authors discuss existing models for the role of the Z-ring and differentiate between a model where either FtsZ generates a force by a conformational change of their filaments or simply provides a scaffold for the assembly of the divisome. What is missing however, is a discussion how the directionality provided by treadmilling could effect the system and how a possible homogenization of PG synthases as suggested by Yang et al. might affect cell wall synthesis. Therefore, I think it would be very informative if the paper discussed these recent findings that helped to suggest a new model for bacterial cell division.

The authors show that constriction requires an inward directed force in combination with PG synthesis, this is a very interesting and important conformation for a hypothesis that exists in the field for a long time. While their simulations show that this force is required to decrease the diameter of the cell, the authors actually do not show if complete division or the formation of two new spherical poles would be possible as the simulations stop after a constriction of only 10 nm.

To me, it looks as the mechanism the authors propose would lead to a double-cone like structure at the division site rather than two half-spheres, similar to this >< rather than). I think this is important, as it would mean that 1. the ends of the in silico cells would accommodate much more PG than real cells actually do. E.g for spherical cell poles the percentage of the cylindrical surface area to the pole surface area is around 25% (Turner et al 2018), what is this value for in silico cell walls? and 2. that there could be another mechanism required to give rise to two half-spheres. Furthermore, the way the data is represented makes it difficult to compare it to in vivo results. Could the authors for example also show plots like Fig 1D with "time" at the x axis in addition to "New/Old PG" or calculate the actual rate of constriction in their simulation? While single molecule rates might not be known, the simulations do give information about measurable parameters. This would make it easier to compare the theoretical results to the rates that are known for cells.

In summary, I would suggest to 1. rewrite the manuscript to clarify if the mechanism shown here can only explain incomplete cell wall constriction or if complete cell division requires additional ingredients 2. better discuss the different protein machineries involved in elongation and division, 3. compare the quantitative values from the simulations with measured in vivo rates, and 4. discuss how directional treadmilling and directed transport of PG synthases as shown by Bisson-Filhon et al and Yang et al might affect the result, 5. Find simulation parameters that would give rise to half-spherical poles.

Reviewers' comments:

Reviewer #1 (Remarks to the Author):

Simulations suggest a constrictive force is require for Gram-negative bacterial division

In this Manuscript, Nguyen et al investigate the mechanisms allowing the constriction of Gram-negative bacterial during division. To this end, the authors rely mostly on simulations of the cell wall mechanics, in combination with analysis of electron microscopy images. They report that the "make before break" mechanism of wall generation does not suffice to create a division furrow, while this is permitted by a constriction force, that could be attributed to FtsZ contraction.

This work is interesting as the septum-less division in Gram-negative bacteria is an interesting feature, contrasting with the usual view from Gram-positive bacteria, yeast, and plants. The manuscript builds from the 2015 article introducing the simulation methods for wall growth, and extends it to study cell division. The exact mechanism of force generation is not within the scope of this manuscript.

The concerns raised by this work are mostly the extreme specificity of the simulation software, and the lack of more general physical considerations.

We agree with the reviewer that the nature of a simulation is that it is exquisitely specific – it is in fact a mathematical calculation – but we did in fact test a broad variety of different general physical considerations. Each of our simulations tested a specific possibility, to see whether it would lead to cell wall constriction. These possibilities were (1) simply concentrating cell wall synthesis at the midcell, (2) including a constrictive force, (3) remodeling the cell wall in a make-before-break mechanism, and (4) a combination of (2) and (3). We chose these because they are the most prominent current conceptual models debated in the field. We also provide the software, and structured it so that the simulations can be re-run with different specific parameters by anyone interested without changing the source code. All the parameters are listed in a single input file.

Specific comments :

1- Simulation

Simulating bacterial cell wall mechanics is no small feat. The exact assumptions of the model, and its implementation, would greatly benefit to be better detailed. Such a detailed description is also sadly incomplete in the 2015 PNAS article.

Thank you. We have now expanded our explanations in the revised version. Please also note that when the first version of our software REMODELER 1 was released (Nguyen et al., PNAS 2015), we described the model including (1) coarse-graining the cell wall (glycan strand and peptide crosslink) and all the equations governing forces, (2) building the initial sacculus model,

and (3) implementing the functions of the enzymes (transglycosylase, transpeptidase, endopeptidase) and their physical constraints. In addition to the Methods section, did the Reviewer see the 10-page Appendix that explains everything in more detail? For every hypothesis implemented, we explained its assumptions and how we implemented it. Further, we posted our source code with the paper. Is the Reviewer aware that we have also now published a 23-page book chapter that describes not only how we built our model but also how we analyzed the data and made the visualization (Nguyen et al., 2016, *Methods Mol. Biol.* 1440, 247–270)?

In the Methods section of the current paper, we describe what aspects of the current model are the same, which are different, and what has been added. The main differences between REMODELER 1 and REMODELER 2 are that the make-before-break mechanism and an external constrictive force have been implemented.

1-A : Physics

- Limitation of steady-state cell radius : except from the initial number of beads per hoop, what sets the cell radius ? If it is indeed only the initial number of beads per hoops, errors in insertion would make the cell diameter unstable : the larger the cell, the more the pressure force ($P R^2$). Over thousands of generations, there must exist a physical or biochemical basis maintaining cell diameter. While the current study addresses a single generation, this is crucial to understand the physics underlying the simulations, and therefore its accuracy.

We agree that there must be forces that regulate cell diameter over generations, and how the process of cell division bypasses them must be very important, but these things are not currently understood. In our previous work we found principles that maintain the rod shape over a few generations (Nguyen et al., *PNAS* 2015), but we admit there was no guarantee that the same principles would maintain the radius over thousands of generations. While it has been proposed that the intrinsic curvature of MreB might be the regulator, it is not clear if MreB forms long filaments in real cells, and if it does, how a curved filament might govern the cell radius. In contrast, a recent work posted on *bioRxiv* by Dion and Garner et al. quite convincingly showed that MreB does not determine the cell radius. While this latest work (Dion et al.) proposed that the cell radius is the result of two competing mechanisms, one expanding cells and the other thinning cells, no detailed molecular mechanism about either mechanism was mentioned. We have now revised our current manuscript to acknowledge and call all this to the reader's attention.

- Strands as lines vs ribbons : in the article, strands are lines of beads. When the authors mention a specific angle $\theta_0=3.14$ between segments, may we safely assume they mean a zero spontaneous curvature for the strand ? Note that for a line, spontaneous curvature is not well defined, it is so only for ribbons, that have an orientation.

Yes, the spontaneous curvature is zero and bending in any direction is penalized. Strand twisting is not considered. While this is only a coarse-grained model, its parameters were chosen to reproduce the behavior of all-atom simulations of a single strand (see Nguyen et al. 2015).

In the model, the crosslinks alternate between each side of the strand. Once again, this amounts as treating the strand as a ribbon rather than a line.

Since adjacent disaccharides rotate roughly 90 deg along the strand (Burge et al., 1977; Labischinski et al., 1979; Kim et al., 2013; Gumbart et al., 2014) and every disaccharide has a stem peptide, the adjacent peptides also rotate 90 deg along the strand, meaning for every peptide pointing left, there is one pointing right, one pointing up, and another pointing down. In our model, we represent two disaccharides as one bead and ignore the peptides pointing up and down. This results in peptides alternating between each side of the strand, but the strands are free to rotate.

As far as I understood, this alternative pattern was set directly in the simulation (by preventing successive binding between two same strands), not derived from an underlying mechanisms. This is a very strong assumption.

Please note that in our previous work (Nguyen et al., 2015) we started with models in which alternation of peptides on the two side of the strand was not applied, allowing successive peptides to form successive crosslinks on the same side. These models resulted in cell bulging since they allowed short segments of one strand to be crosslinked to long segments of another strand. Only after analyzing the results of these models and then taking into consideration the 90-deg rotation of adjacent disaccharides did we start implementing alternation of successive peptides on the two sides of the strand. While this implementation of peptide alternation helps maintain the rod shape, we understand that cells might have unknown mechanisms to fix small bulges preventing them from growing as cell wall synthesis continues. Regardless of specific molecular mechanism, the alternating pattern of crosslinks must be maintained to maintain cell shape over many generations of growth and division. That all being said, we have now revised the manuscript to highlight this assumption and the reason we include it.

How much does this affect the stability of the cell radius ? Even a few errors would likely create the cell wall to be unstable at some timescale (see first point).

Yes, crosslink alternation on the two side of the strand is important to maintain cell radius. As mentioned above, we showed in our previous work cases where a few small bulges caused by errors in crosslink alternation eventually accumulated and caused shape loss.

- Length of strands and speed of enzyme translocation is also given by simulation assumptions and does not come from physical or biochemical principles.

Yes. Unfortunately the rates of the enzymes in cells are not known, so we had to use estimates. As pointed out in our previous paper (Nguyen et al 2015), however, we chose enzyme rates in a way that reproduced the average length of glycan strand within the range of 21-33 disaccharides,

which was reported experimentally (Glauner et al., 1988; Harz et al., 1990). Readers will be able to test other rates easily with our provided software if future experiments reveal them. That being said, we maintain that the main conclusions of our paper are intuitive, once visualized and explained, and are not dependent on these details.

This could be problematic if there were feedbacks between strand length and mechanics. Here this should not play too large of a role, but it would be useful to confirm.

Correct! We have tested this, and strand length does have a small effect on the cell radius. Since turgor pressure expands the cell, pulling glycan strands circumferentially, this also stretches peptide crosslinks, pulling those that connect strand termini away from (not parallel to) the cell axis. The hoop circumference is therefore the sum of the total length of the strands that make up the hoop plus the length of the terminal crosslinks between them. For hoops composed of the same number of disaccharides, as the average strand length increases, the number of terminal crosslinks is decreased making the circumference smaller (see Figure S1, copied below), but the effect diminishes gradually as strand length becomes longer and longer. We have now added all this to our revised manuscript.

Figure S1: Radius of the cell wall cylinder under a turgor pressure of 3 atm. The cell wall was made of hoops with the hoop size of 400 tetrasaccharides. Increasing the average PG length reduced the number of terminal crosslinks therefore reducing the cell wall's radius.

- The authors assume that strand extension depends on crosslinking. If I understood correctly, extension is 3000x more likely after crosslinking. This is a crucial assumption both for the stability of cell radius and the make before break mechanism.

We assume the reviewer is referring to the hypothesis that transpeptidation facilitates translocation of transglycosylase. To make it clear, strand elongation is modeled in three

consecutive steps, each with their own rates: (1) the transglycosylase loads a precursor bead, (2) the loaded precursor is linked to the strand tip, and (3) the enzyme translocates to the new strand tip for the next strand elongation event. The mentioned hypothesis changes the rate of step 3 (*translocation*) according to the crosslinking state: it occurs once every $2 \cdot 10^6$ simulation time steps if the strand tip is not crosslinked, and once every $3 \cdot 10^3$ time steps otherwise, meaning the *translocation* rate is $\sim 700x$ (not $3000x$) more likely after crosslinking. As shown in our previous work (Nguyen et al., 2015), in early models where there wasn't a rule regulating strand elongation, many long, un-crosslinked strands were created that later bent and distorted the cell morphology. We therefore added this hypothesis as a regulation of strand elongation. Regardless of whether a make-before-break mechanism is active or not, the presence of long, uncrosslinked strands leads to shape defects and some kind of regulation of strand elongation is needed. We understand, however, that ours is only one hypothetical mechanism of regulation, and the one that actually acts in cells is not known. That being said, the revelations of our simulations about why make-before-break is insufficient to drive cell constriction by itself do not depend on strand extension rates.

1-B : Simulations

- Compilation : the provided code could not be compiled out of the box as suggested in the readme. I had to fix an overlong line (132 characters), replace the instruction `-openmp` by `fopenmp` and add the instruction `-fno-check-range` (gfortran on Ubuntu 16LTS). It then compiled and run on 4 cores in parallel.

Thank you for finding this incompatibility, which we did not experience when using Intel Fortran. We have now fixed the 132-char line and updated the readme to address this problem.

- Open source code : the authors provide the source code of remodeler2 which is a great choice. The parameter file has a simple syntax and the parameters can be modified easily.

Thanks!

Due to limited commenting, modularity (limited partly by the choice of Fortran90) and documentation, it should not be considered as a large-purpose platform that would be widely used and built upon by the community. Rather, it is a program fulfilling a specific simulation task in a fast and efficient way.

We agree. As the reviewer pointed out, simulation of this system is not a small feat. Bacterial cell biology is a long-term interest of our group. We have contributed important data through electron cryotomography about cell wall architecture and the presence/absence and structure of cytoskeletal filaments. This paper is one of a series that first addressed cell elongation, and now explores concepts that might drive constriction, and we have another in the works on how FtsZ filaments might provide constrictive force. We are still obviously at the stage of exploring general ideas about basic, fundamentally different mechanisms that MIGHT be in play in real cells, in as efficient way as we can. That is why we have not yet invested the additional effort to

make the software modular and extensible by all. That will come later once we know which ideas deserve greater attention.

2- Constriction force

The constriction force per unit length (a tension, maybe this term could be used to avoid confusion)

FYI we did not call it tension, but simply the constrictive force per unit length of circumference for a reason. Similar to tension, the constrictive force that is necessary for constriction also scales with both turgor pressure P and the cell radius R ($\sim P \cdot R$), but the constrictive force divided by the circumference scales only with P . Therefore, we for now stick with calling it “the constrictive force divided by circumference”.

necessary is of the order of 20pN/nm. The buckling tension is of the order of 35pN/nm. It is important to highlight that these values are expected from scaling arguments (tension $\sim P \cdot R$) and should not be overly model dependent.

Yes - the buckling “constrictive force divided by the circumference” scales with P , but not R , while the buckling constrictive force itself scales with both P and R ($\sim P \cdot R$).

This scaling with size and pressure could be confirmed by simulations ; the numerical cost of simulations (not discussed in the article) might be a limiting factor for this systematic approach.

Agreed! Computational resources (several weeks/simulation) prevented us from systematically simulating many different cell sizes. However, as the reviewer points out, a simple physical consideration can already point out the buckling force scales with pressure and size. We appreciate the point and have now added it to the revised text for all readers.

The tension of 20pN/nm, i.e. a global force of 15nN is the reason why other turgid walled cell need to form a septum. The absence of a such septum is the crucial point addressed by this article. The conclusion that make before break (MBB) cannot work because of the wall tension caused by pressure is very important.

However, the authors quickly dismiss the hypothesis that would allow MBB to work, namely the existence of a pre-stress. For instance, a pre-stress has been reported for spectrin in red blood cells (Turlier et al. Nat Phys 2016). Could FtsZ force cause a such pre-stress ? What value should we expect for a pre-stress, so that MBB works ? It is likely that this value is similar to the constriction tension.

We assume the reviewer is referring to pre-stretching PG strands before they are incorporated into the wall. We agree that if new strands are pre-stretched to the same degree that existing strands are, MBB might work. We have revised the text to make it clear that a pre-stretching

mechanism might enable the make-before-break without the need for a constrictive force. Please also note that new-PG pre-stretching was one of the three conceptual models for cell wall division that we discussed in our previous manuscript.

We were at a loss, however, to imagine a scenario where a constrictive force that is generated in the cytoplasm could be translated to a stretching force in the periplasm where the new PG is. Nevertheless, we understand that we can never imagine all possibilities! So the answer is Yes, FtsZ might be able to cause pre-stretching, but we could not imagine how, so we did not simulate it, though we state clearly in our manuscript that we have no evidence refuting this model.

Lastly, a tension of ~ 20 pN/nm matches, or is beyond, the usual estimate of membrane-breaking tension. What prevents here the membrane from (a) peeling of the wall (b) rupturing ?

We believe the turgor pressure of the cell is mostly sustained by the cell wall behind the inner membrane. A constrictive force might then simply reduce turgor pressure at mid-cell, allowing the cell wall to relax slightly (from an otherwise stretched state). The net force (constrictive force plus turgor pressure) on the membrane would be small.

Reviewer #2 (Remarks to the Author):

This study builds on the authors' 2015 paper, which derived a computer model for PG assembly in Gram-negative bacteria. That model, based on a PG wall that is one layer thick, successfully predicted elongation of the PG wall, avoiding bulges and ruptures and maintaining a constant diameter over generations. Now the authors adapt their model to try to explain how PG remodeling can achieve constriction during cell division. This addresses a current controversy in the field of bacterial cell division: can PG remodeling provide a constriction force by pushing the membrane from the outside, or is the constriction force provided primarily by FtsZ, with PG remodeling just accommodating constriction? The authors try several schemes for insertion of new strands, and conclude that two mechanisms can accommodate constriction of the PG wall. A major conclusion is that constriction against turgor only occurs when there is an external force that can constrict the sacculus to a radius less than that of a relaxed (no turgor) state. Therefore PG remodeling is not driving constriction, but simply accommodates constriction driven by the external force, likely FtsZ. This will not be the last word on the controversy – its conclusions are limited to the assumptions of the model. But advocates of PG pushing will be challenged to come up with a molecular mechanism, perhaps with modified rules for the network of glycan strands and peptide crosslinks and the insertion of new ones. I recommend publication after attention to some questions and concerns.

We completely agree that our work is not to end the discussion, but rather an initiation of a new phase of the discussion based on mechanistic simulation. It reports the molecular consequences of the major ideas being debated in the field, given the specific assumptions and parameters of the simulation.

1. The model is based on a PG wall limited to one layer. Is there evidence that the PG is really only one layer, as opposed to two or three layers?

In our previous electron cryotomography work on purified sacculi of *E. coli* and *C. crescentus*, we showed that the sacculi was a single-layer thick where it folded over on itself (it collapsed into a flattened balloon-like shape in the thin ice). We therefore believe the PG is mostly one-layer, but it is possible it was affected by the sample preparation, and we do not know whether there are patches of dual- or multilayer wall. Note in this manuscript we show tomograms of dividing cells, where it is clear that the PG is not noticeably thicker in and around the division plane than elsewhere. That is the data on the subject we have at this time.

That being said, we believe that the principles we found with our simulations would hold true even for a PG model a few layers thick. For example, as discussed in our manuscript, for an *E. coli* cell to divide via a make-before-break mechanism that builds a septum of several PG layers, its wall hoop size of ~ 3000 disaccharides would mean the septum would have to be ~65 nm thick for the innermost PG layer to be relaxed enough that the make-before-break mechanism would work without a constrictive force. This is of course far thicker than those seen in the cryotomograms. Nevertheless, we have now modified our manuscript to make it clear that our findings are limited to models of a single-layer PG.

2. The cryotomograms are introduced in the middle of the section “make-before-break mechanism, interrupting the flow of that section.” I would suggest moving the cryoEM to a separate section at the end “Can Gram-negative bacteria divide by a septation mechanism.” Here you might expand a bit and explain that you are addressing Erickson’s (2017 Bioessays) suggestion, that a septation mechanism could allow cell division to cheat turgor pressure by building a new septum inside a periplasm that was isoosmolar with the cytoplasm. However, this mechanism requires at least a vestigial septum protruding from the cell wall into the periplasm. Erickson suggested that this might exist based on published cryotomograms. In contrast, you are now presenting an expanded gallery of new cryotomograms that show no hint of a septum at your 4 nm resolution. You might clarify that this failure to see a septum is presented as a contradiction of his hypothesis, as applied to Gram-negative bacteria.

Thank you for these insights. We have modified the text as suggested.

3. Keep in mind, however, that Gram-negative bacteria can grow PG septa under some conditions (Erickson, 2017; Uehara et al, J. Bacteriol. 191:5094). This suggests that the purely constrictive division seen in these cryotomograms may share mechanisms with septal division. This would require a more complex system than the present model.

Agreed. We have added a note that septation was observed in Gram-negative bacteria in some abnormal conditions.

4. Fig. 3 is difficult to interpret. The text speaks of beads, which are not on the Fig. From the previous paper I believe beads should be at the intersection of each peptide with its glycan.

Sorry for this confusion. Yes, there is a glycan bead at each intersection of each peptide. We thought removing the beads and just showing the bonds would make the view clearer, but apparently this did not work. We have now added the beads back to the visualizations.

It seems that the cyan target peptide is changed from B to C: the one in C is now on the left and is a bit further down. It may help to call attention to this detail.

Correct! As the constrictive force increased, it pulled the new strand tips down further and changed the target peptide. To make it clear, both the old peptide target and the new one are now highlighted, the old still in cyan and the new now in yellow. We also modified the figure legend to describe this.

Also in Fig. 6 I could not find the highlighted strands. If they are orange I can't distinguish them from red.

Sorry the visualization was not clear. In fact, the highlighted crosslinks are in cyan. We have now increased the size of the highlighted crosslinks and added arrow heads pointing at them to make them stand out better in Fig. 6D-G. We have modified the figure legend accordingly.

5. In Discussion the authors write “(2) If a mechanism exists to initially compress existing hoops ..(3) If a mechanism exists to initially relax existing hoops.” I think compress and relax may refer to the same thing: constricting to the same state as a relaxed sacculus? Do clarify.

Sorry for the confusion. We meant in (2) the hoops are compressed into a state that are *smaller* than relaxed hoops, i.e., they are beyond relaxed, they are compressed. In (3), the hoops are just relaxed. We have modified the text to clarify.

6. I think the “make before break” mechanism is described in a confusing order (apart from the interruption by the cryotomograms). I suggest first outlining how it might work, by the forward skipping mechanism in Fig. S3. Then explain that this would actually work without turgor. Then explain that turgor increases the gaps between glycans, and this defeats the skipping mechanism.

Thanks you – we had our reasons for the order, but we have now modified the text to follow this suggestion.

Am I correct that the defeat is primarily due to the average length of the glycan strand being about the same as the distance needed for a skip.

We're not exactly sure what you mean, so let us clarify: MBB alone fails because as the new strand is inserted, because it is at lower radius, disaccharide by disaccharide, it gradually "gets ahead" of the disaccharides in the old strand above it. But because gaps arise frequently in the hoops, the new strand is almost never far enough ahead to skip a gap, and therefore the progress is lost, and new hoops come out with the same number of total disaccharides as the old hoops, and when turgor pressure hits them, they expand, and no constriction results. Now with numbers: in our simulated model with $N_h=400$ tetrasaccharides per hoop, turgor pressure expands the radius by $\Delta R \sim 10$ nm, therefore adding 62.8 nm to the circumference, almost all by reorienting the peptide crosslinks across gaps to perpendicular to the cell axis. As the average length of each glycan strand is $N_s=14$ tetrasaccharides, there are on average $N_h/N_s \sim 29$ gaps around the hoop. Thus the additional circumference in each gap is $2\pi\Delta RN_s/N_h \sim 2.2$ nm. Meanwhile the new strand, added below (slightly smaller radius), again with average length of $N_s=14$ tetrasaccharides, only gets $2\pi\Delta r N_s/N_h = 0.44$ nm ahead of the old strand before the strand terminates, where $\Delta r \sim 2$ nm is the thickness of a single PG layer. Thus even in our extra-small-sized simulated sacculus, the glycan strands are too short to get far enough ahead of the old strands to skip a gap. For a cell of an *E. coli*'s size ($N_h \sim 1500$ tetrasaccharides per hoop), turgor pressure expands the radius by $\Delta R \sim 65$ nm (Fig. S4), adding $\sim 2\pi\Delta RN_s/N_h \sim 3.8$ nm to each gap and a strand length of $N_s = 14$ tetrasaccharides only gets $2\pi\Delta r N_s/N_h \sim 0.12$ nm ahead of the older strand before it terminates, much too little to skip the gap and eliminate a disaccharide from the total hoop.

7. A recent paper from Rojas..Huang et al (Nature 2018) showed that the outer membrane contributed substantially to the mechanical rigidity of the cell wall. Does this affect your model?

No. While the finding that the outer membrane has a comparable stiffness to that of the cell wall was interesting, Rojas et al. also found that *the outer membrane bears little or no load in the exponential growth phase*, since the surface area of the cell did not change when turgor pressure and the cell wall were eliminated. This is all consistent with the assumption of our simulations that the cell wall bears the turgor pressure. In any case, the principles we discovered through simulation about the consequences of MBB or a constrictive force will be true regardless of how stiff the outer membrane is.

8. The forces you find necessary are very large, much larger than those previously suggested by Lan...Sun (2007). Can you discuss how your models differ from Sun's?

Lan et al. PNAS (2007) assumed new PG was pre-stretched 20% before being incorporated. This brings the new PG to a similar stretching degree of the existing PG. In our model, we do not assume that new PG is pre-stretched, so the constriction force needs to constrict the cell wall $\sim 10\%$ to bring the existing cell wall to a relaxed state similar to that of the new PG. Note also that the change in stretching degree in our model was smaller than that in Lan et al.'s, but this might be due to the fact that we chose to simulate a miniature cell. The model by Lan et al. PNAS (2007) and ours are therefore consistent with one another on the point that to constrict the cell wall, new PG and existing PG must be brought to a similar stretching condition so that they are either stretched (Lan et al.) or relaxed (ours). Since in Lan et al.'s model pre-stretching

already generates a favorable condition for constriction, the magnitude of the constriction force required is small. While we discussed new-PG pre-stretching as one of the three conceptual models of cell wall constriction, we have not yet explored it in detail through simulation. We have now clarified these points in the revised manuscript and cited Lan et al.'s for having the same assumption.

Also, the forces given here are for the 137 nm radius. If you move to the 500 nm radius of *E. coli*, do the forces get larger or smaller?

Since tension is proportional to the radius, the force will get proportionally larger.

Reviewer #3 (Remarks to the Author):

This paper addresses one of the most important questions in bacterial cell biology: how do cells generate the forces required to divide? Currently, there are two schools of thought to explain the mechanism of bacterial cell division: first, that the bacterial tubulin FtsZ is the main force generator providing the an inward directed force required for the cell to constrict and split into two and second, that the force originates from the other side of the membrane, namely by the cell wall that is growing inward. According to the later model, the cytoskeletal ring provided by FtsZ would only act as scaffold guiding the inward growth of the cell septum. Using a very elegant a powerful previously established computer simulation approach, the authors of this paper were able to test both models "in silico". Their study nicely and convincingly shows that for cell constriction a combination of both is most probably applied: inward growth by a make-before-break-mechanism supported by an inward pulling force compensating for the turgor pressure.

Actually, just to clarify, we identify two possibilities: (i) make-before-break + a mild constriction force, or (ii) a stronger constriction force alone (without any special modifications to the cell wall synthesis process).

The claims of the paper are of interest for many other researcher in the community, even though the hypotheses made have existed for a long time and the findings are probably not too surprising. Nevertheless it will help to motivate and design future experimental work.

We appreciate the Reviewer's support and feedback, but we believe our work reveals that when carefully considered in molecular detail (for the first time), the make-before-break mechanism is not sufficient to drive constriction. A constrictive force, on the other hand, is. Thus while not proposing any fundamentally new hypotheses, we have given important reasons to favor one over the other.

Below are some concerns I have with the paper, I hope they are helpful to either clarify or improve the manuscript.

The main assumption of this study and its model is that cell wall synthesis does not work fundamentally different during elongation and division with the only difference being an inward directed force during cell division presumably provided by the Z-ring. The author's provide three reasons to justify this assumption, however, none of them is actually very convincing and would need a more detailed explanation .

We understand that this is an assumption. To assist the discussion, our three reasons are (a) some overlapping roles of the enzymes, (b) similar movements of some proteins, and (c) similar orientation of glycan strands. The mode of PG synthesis might however still be very different between cell elongation and cell division. We have now called attention to this assumption in the Discussion and pointed out that it might be wrong.

1. Enzymes involved in elongation and division have overlapping roles: While this is true for PBP1B and PBP1A, other proteins are essential and only perform one task, such as PBP2 (elongation) and PBP3 (cell division). It is true that they are structurally and functionally related to other PG synthases, however, they are part of very different complexes and subject to different regulation. Therefore cell division complexes might function very differently than elongation complexes and could give rise to a different connectivity of the PG network. This was also pointed out by Turner et al 2018 cited in this paper: “the poles and the cylindrical part of the cell are biosynthesised by different sets of nano-machinery”

Very good point. We have revised our text to make it clear that only PBP1A and 1B have some overlapping role, but PBP2 and PBP3 are part of two different complexes (MreB-associated complex and FtsZ-associated complex, respectively) and that while we assume these two complexes work in a similar mode, other possibilities exist.

2. PG synthesis proteins were shown to move around the cell's circumference during both elongation and division: This over-simplifies what we currently know about the intracellular dynamics of PG synthases and their activities. First, in the case of E. coli elongation, directional movement was only found for filaments of MreB, which was dependent on the elongation specific TPase PBP2, not for the division specific TPase PBP3 or GT-TPases PBP1A and PBP1B. In contrast to FtsZ, which itself treadmills to drive the motion of PBP3. Accordingly, the PG synthase complex are either the driver (elongation) or the passenger (division) of the observed motion, which suggests that cell wall synthesis for both processes could be very different. In fact, it is currently not known if during septal PG synthesis the corresponding enzymes are moving (as PBP3), static (in complex with a cytoskeletal element) or diffusing (freely diffusing, see Cho et al 2017). Therefore, the situation is much more complex and should be appropriately discussed.

Agreed again. We are of course aware of the data by Yang et al., 2017 that showed that the speed of PBP3 depended on the treadmilling speed of FtsZ. In our opinion, this does not necessarily mean that PBP3 is simply a passenger. Just like PBP2 being a driver dragging MreB forward, PBP3 could still be a driver itself while its speed depends on the rate of FtsZ-mediated fuel (PG precursors) supply. If a complex is attached to specific monomers of a treadmilling filament, they will of course not advance. Some cellular objects like plasmids (and even whole cells like listeria) are pushed forward by localizing to the growing tips of a *bundle* of filaments. As long as the filaments take turn to release the passenger and add a subunit to their tips, the passenger still hang on the growing bundle. However, we have never seen bundles of FtsZ filaments terminating at the same location in our tomograms and therefore judge that the situation is different in cell division. To hang on the growing tip of a single FtsZ filament after unbinding from the old tip, the “passenger” would have to *actively translocate* to the new one, meaning acting as a *driver*. We have added a discussion of these points to the revised paper.

3. The orientation of glycan strands is similar at the poles and cylindrical parts of E. coli: this was nicely shown experimentally by Turner et al 2018, but even if the orientation of strands is similar, the length distribution might be very different.

Agreed. We have revised our text to acknowledge this point.

Next, the authors discuss existing models for the role of the Z-ring and differentiate between a model where either FtsZ generates a force by a conformational change of their filaments or simply provides a scaffold for the assembly of the divisome. What is missing however, is a discussion how the directionality provided by treadmilling could effect the system and how a possible homogenization of PG synthases as suggested by Yang et al. might affect cell wall synthesis.

To be clear, when referring to the model of FtsZ that serves as a scaffold for the cell wall synthesis machinery we did not strictly mean a “static scaffold”. In light of the recent results by Bission-Filho et al., 2017 and Yang et al., 2017 that showed FtsZ treadmills around the cell, the scaffold model should be understood as a “moving/treadmilling scaffold”. We have now cited the two papers that showed FtsZ treadmilling and mentioned their speculations on how treadmilling might help distribute PG synthesis evenly around the cell.

We could not, however, imagine a conceptual model in which the directionality provided by treadmilling of FtsZ might affect generation of a constrictive force and therefore did not speculate this possibility, but we are open to further suggestions. Instead, just like Bission-Filho et al. and Yang et al. speculated, we could only imagine treadmilling might have something to do with distributing PG precursors around the cell circumference since the enzymes (or some of them) need to move around to make circumferential glycan strands. While during cell elongation, MreB, which doesn't need to form long filaments since its likely function is to simply supply PG precursors to the circumferentially-moving enzymes, can be easily dragged along due to its light weight, the situation might be a little different for FtsZ during cell division. While supplying PG precursors might not require FtsZ to be a long filament, generating a

constrictive force might for efficiency. The drag of long filaments could prevent the movement of enzymes if they were attached. In this case, treadmilling helps solve the problem.

We think the reviewer's "possible homogenization of PG synthases as suggested by Yang et al." referred to the last sentence of Yang et al.'s paper: "*The broad conceptual similarities among FtsZ, MreB, and the movement of cellulose synthase complexes along cortical microtubules in plants (36) suggest that coupling cytoskeletal motion to wall synthesis may be a general strategy across the kingdoms of life to ensure evenly distributed, robust septal wall synthesis and morphogenesis through time-averaging.*" As mentioned above, we agree with this speculation by Yang et al. that coupling between cytoskeletal motion and cell wall synthesis ensures a homogeneous distribution of the new material.

Therefore, I think it would be very informative if the paper discussed these recent findings that helped to suggest a new model for bacterial cell division.

We have now revised our text to discuss treadmilling as helping to ensure homogeneous distribution of new PG around the circumference. Again we maintain that the principles our simulations revealed about make-before-break and constrictive forces are both interesting and valid regardless of if or how FtsZ might distribute cell wall synthesis evenly around the ring.

The authors show that constriction requires an inward directed force in combination with PG synthesis, this is a very interesting and important conformation for a hypothesis that exists in the field for a long time. While their simulations show that this force is required to decrease the diameter of the cell, the authors actually do not show if complete division or the formation of two new spherical poles would be possible as the simulations stop after a constriction of only 10 nm. To me, it looks as the mechanism the authors propose would lead to a double-cone like structure at the division site rather than two half-spheres, similar to this \gg rather than \gg). I think this is important, as it would mean that 1. the ends of the in silico cells would accommodate much more PG than real cells actually do. E.g for spherical cell poles the percentage of the cylindrical surface area to the pole surface area is around 25% (Turner et al 2018), what is this value for in silico cell walls?
and 2. that there could be another mechanism required to give rise to two half-spheres.

Unfortunately we cannot address these issues well at this time because even our longest simulations so far (constricting only ~30% of the original diameter) required up to a month of wall-clock time. In this first phase of the project, we were exploring widely different ideas, with different parameters, and so no single simulation could be run for months and months. We hope to further refine the models and code in future phases of the project and run them deeper into constriction.

Furthermore, the way the data is represented makes it difficult to compare it to in vivo results. Could the authors for example also show plots like Fig 1D with "time" at the x axis in addition to "New/Old PG" or calculate the actual rate of constriction in their simulation? While single

molecule rates might not be known, the simulations do give information about measurable parameters. This would make it easier to compare the theoretical results to the rates that are known for cells.

We are unable at this time to convert simulation time steps to actual time in cells, since we really don't know the enzyme activities in vivo. We believe we have reported the specifics that we can at this time. Note also that we modeled a miniature cell, and constriction rates very likely depend heavily on constriction stage, and are different in different species, in different growth conditions. In response to this comment, however, we have now emphasized in the Discussion that we do not believe our model is complete in representing all the factors involved in cell division in any particular bacterial species, and our findings are limited to our assumptions, which are based on the current literature. That being said, the simulations reveal for the first time the effects of different hypothesized mechanisms at the level of individual molecules and hoops in a sacculus.

In summary, I would suggest to 1. rewrite the manuscript to clarify if the mechanism shown here can only explain incomplete cell wall constriction or if complete cell division requires additional ingredients 2. better discuss the different protein machineries involved in elongation and division, 3. compare the quantitative values from the simulations with measured in vivo rates, and 4. discuss how directional treadmilling and directed transport of PG synthases as shown by Bisson-Filhon et al and Yang et al might affect the result, 5. Find simulation parameters that would give rise to half-spherical poles.

Thank you very much for all helpful discussion and suggestions. We have done 1, 2, and 4, but must leave 3 and 5 to future studies.

REVIEWERS' COMMENTS:

Reviewer #1 (Remarks to the Author):

First, I thank the authors for their instructive answers.

The changes clarify several points in the manuscript as well as highlight the limitations of their approach. The manuscript gives interesting results, and proposes a powerful tool, on an important question : how do bacterias divide.

While I'm still concerned by some of these limitations (long term maintenance of diameter, and possibility of PG pre-stress), these are due to the current state of the knowledge. I agree that the work in this manuscript will help address these limitations in the future. I therefore recommend the manuscript for publication.

Minor points :

-In turgor pressure, how is the gradient of the volume computed ?

-In Langevin dynamics : since the authors are using overdamped dynamics, they might as well present the overdamped equation. They should in any case mention the numerical scheme used to integrate the equation.

Additional point, *not relevant to publication* :

"We are still obviously at the stage of exploring general ideas about basic, fundamentally different mechanisms that MIGHT be in play in real cells, in as efficient way as we can. That is why we have not yet invested the additional effort to make the software modular and extensible by all. That will come later once we know which ideas deserve greater attention. "

I personally disagree with that statement, as I think it is exactly the right time to have a modular code, so that the community may test their own ideas, rather than only be able to change parameters. If the community is large enough, you would benefit from the input of other developers.

Reviewer #3 (Remarks to the Author):

The authors have address all my concerns, this is a very interesting paper that should be published.

REVIEWERS' COMMENTS:

Reviewer #1 (Remarks to the Author):

First, I thank the authors for their instructive answers.

The changes clarify several points in the manuscript as well as highlight the limitations of their approach. The manuscript gives interesting results, and proposes a powerful tool, on an important question : how do bacterias divide.

While I'm still concerned by some of these limitations (long term maintenance of diameter, and possibility of PG pre-stress), these are due to the current state of the knowledge. I agree that the work in this manuscript will help address these limitations in the future. I therefore recommend the manuscript for publication.

Thanks.

Minor points :

-In turgor pressure, how is the gradient of the volume computed ?

As the volume inside the wall was divided into a set of tetrahedrons, the gradient of the volume was the sum of gradient of individual tetrahedrons. Therefore the force on bead i will be the sum of gradients of the tetrahedrons that contain bead i as a vertex. We have added to the text the description of this step.

-In Langevin dynamics : since the authors are using overdamped dynamics, they might as well present the overdamped equation. They should in any case mention the numerical scheme used to integrate the equation.

We have added the overdamped equation and numerical scheme to the Methods section.

Additional point, *not relevant to publication* :

"We are still obviously at the stage of exploring general ideas about basic, fundamentally different mechanisms that MIGHT be in play in real cells, in as efficient way as we can. That is why we have not yet invested the additional effort to make the software modular and extensible by all. That will come later once we know which ideas deserve greater attention. "

I personally disagree with that statement, as I think it is exactly the right time to have a modular code, so that the community may test their own ideas, rather than only be able to change parameters. If the community is large enough, you would benefit from the input of other developers.

We take note of this. Thank you.

Reviewer #3 (Remarks to the Author):

The authors have address all my concerns, this is a very interesting paper that should be published.

Thank you.